# Iterative Missing Data Imputation with Model Form Adaptation and Non-Missing Feature Supervision

**Hao Wang**[1][*]  **Zhengnan Li**[1][*]  **Zhichao Chen**[2]  **Xu Chen**[3][†]  **Shuting He**[4]
**Guangyi Liu**[5]  **Haoxuan Li**[6][†]  **Zhouchen Lin**[2,7,8,†]

[1]Xiaohongshu Inc.
[2]State Key Lab of General AI, School of Intelligence Science and Technology, Peking University
[3]Gaoling School of Artificial Intelligence, Renmin University of China
[4]School of Computing and Artificial Intelligence, Shanghai University of Finance and Economics
[5]Department of Control Science and Engineering, Zhejiang University
[6]Center for Data Science, Peking University
[7]Institute for Artificial Intelligence, Peking University
[8]Pazhou Laboratory (Huangpu), Guangzhou, Guangdong, China

## Abstract

Iterative imputation is a prevalent method for missing data imputation, where each feature is imputed iteratively by treating it as a target variable estimated from all other features. However, iterative imputation method suffers from two principal limitations: ❶ **it imposes a single parametric model form to impute all features**, neglecting the potential for optimal models to vary among features, which risks model misspecification; and ❷ **it assumes every feature contains missing values**, overlooking the potential presence of non-missing features, termed as **oracle features**, which are informative for imputation. To address these limitations, we propose kernel point imputation (KPI), a bi-level optimization framework for iterative missing data imputation. At the inner level, KPI adaptively learns the optimal model form for each feature within a reproducing kernel Hilbert space, addressing limitation ❶. At the outer level, KPI utilizes oracle features as supervisory signals to iteratively refine the imputations, addressing limitation ❷. Experiments demonstrate that KPI outperforms competitive imputation methods. Code is available at https://github.com/FMLYD/kpi.git.

## 1 Introduction

Missing data are ubiquitous in real-world data collection and analytics [25, 46, 39]. For example, in manufacturing, temperature sensors may fail due to overheating or electrical disruptions, compromising data integrity and impeding analytical workflows [1]. Similarly, equipment-monitoring systems can experience connectivity loss in electrical sensors, impeding fault detection and introducing security risks [36, 35]. These issues highlight the importance of missing data imputation (MDI) techniques, which aim to recover missing data using observed data, thereby enhancing the integrity of collected datasets [40, 38] and the reliability of subsequent data-driven applications [3, 2, 33, 43].

Existing MDI methods can be broadly categorized as discriminative or generative [7]. On the one hand, discriminative methods, such as statistical imputation (e.g., mean and median imputation [45]) and iterative imputation (which iteratively estimates missing values using univariate models [27, 30]), have been well developed. On the other hand, generative methods have recently attracted attention

---

[*]This work was done in the internship at Xiaohongshu Inc. Both authors have equal contribution.
[†]Corresponding author.

for their capacity to model complex data structures [1]. However, they often encounter training challenges [19, 26] or rely on strong data assumptions [42, 28]. Empirically, generative methods are frequently outperformed by discriminative methods [46, 24]. Therefore, discriminative methods remain the preferred choice for MDI in practice [38].

Among discriminative methods, iterative imputation is widespread for its plain implementation and strong performance. This approach specifies a univariate model for each feature conditioned on the rest and iteratively updates missing values until convergence [27]. Despite its popularity, standard iterative imputation faces two significant limitations. ❶ **it imposes a single parametric model form to impute all features**, which risks model misspecification, as features often exhibit heterogeneous dependencies that cannot be adequately captured by a fixed-form parametric model [7]. ❷ **it assumes every feature contains missing values**, which neglects the potential of **oracle features**, i.e., non-missing features, to serve as high-quality supervisory signals for imputation.

To counteract these two limitations, we reformulate iterative imputation as a bi-level optimization problem. To address limitation ❶, the inner-level optimization adaptively selects functional forms from a reproducing kernel Hilbert spaces (RKHS) for each feature, reducing model misspecification. To address limitation ❷, the outer-level optimization aligns the imputed values with the oracle features, leveraging them as direct supervisory signals. Subsequently, we propose *kernel point imputation* (KPI), which expresses the optimal model as a linear combination of kernel functions, enabling efficient solution via stochastic gradient descent. Furthermore, we design an adaptive kernel ensemble strategy to dynamically combine kernels, thereby enhancing model expressiveness and alleviating hyperparameter selection challenge amidst incomplete data.

**Contributions.** The key contributions of this study are summarized as follows:

- We identify two critical limitations in current iterative imputation methods: ❶ **it imposes a single parametric model form to impute all features**, which risks model misspecification, and ❷ **it assumes every feature contains missing values**, which overlooks the utility of oracle features.

- We introduce KPI, a bi-level MDI framework that addresses the identified limitations while remaining compatible with gradient-based optimizers. Additionally, we develop a kernel ensemble strategy to address the difficulty of hyperparameter selection amidst incomplete data.

- We conduct extensive experiments to demonstrate the superiority of KPI over competitive baseline methods. Our results also confirm that both the model form optimization and the utilization of oracle features can effectively enhance imputation performance.

## 2 Preliminaries

This study focuses on the MDI problem as an *end* goal. We do not treat imputation as a preprocessing step for downstream tasks [7], such as time-series forecasting [33] or pseudo-labeling [12], as these settings typically require joint optimization with task-specific objectives [17].

The MDI problem is specified by the following core components: ❶ **Ideal matrix** $\mathbf{X}^{(\mathrm{id})} \in \mathbb{R}^{\mathrm{N} \times \mathrm{D}}$: the non-missing (ideal) data matrix with N samples and D features; ❷ **Missingness matrix** $\mathbf{M} \in \mathbb{R}^{\mathrm{N} \times \mathrm{D}}$: a binary matrix where $\mathbf{M}_{n,d} = 1$ indicates missingness and $\mathbf{M}_{n,d} = 0$ otherwise; ❸ **Observed matrix** $\mathbf{X}^{(\mathrm{obs})} \in \mathbb{R}^{\mathrm{N} \times \mathrm{D}}$: the data matrix containing missing values, expressed as $\mathbf{X}^{(\mathrm{obs})} = \mathbf{X}^{(\mathrm{id})} \odot (1 - \mathbf{M}) + \mathrm{nan} \odot \mathbf{M}$; ❹ **Imputation matrix** $\mathbf{X}^{(\mathrm{imp})} \in \mathbb{R}^{\mathrm{N} \times \mathrm{D}}$: the matrix that replaces the missing values in $\mathbf{X}^{(\mathrm{obs})}$ with imputed values. On this basis, the task of MDI is to construct an imputation matrix $\mathbf{X}^{(\mathrm{imp})}$ based on $\mathbf{X}^{(\mathrm{obs})}$ that closely approximates $\mathbf{X}^{(\mathrm{id})}$. Different imputation methods differ in how they map $\mathbf{X}^{(\mathrm{obs})}$ to $\mathbf{X}^{(\mathrm{imp})}$.

A prevalent family of methods is iterative imputation, which iteratively treats each feature as a target variable and estimate its missing values from the remaining features. In the training stage, for arbitrary $1 \leq d \leq \mathrm{D}$, define the target feature $\mathbf{Y}_d^{(\mathrm{obs})} = \mathbf{X}_{\cdot,d}^{(\mathrm{obs})}$. The method fits an imputation model $f_\theta$ that learns the dependency between the target feature and the other features by solving

$$\min_{\theta} \left\| \mathbf{Y}_d^{(\mathrm{obs})} - f_\theta(\mathbf{X}_{\cdot,-d}^{(\mathrm{obs})}) \right\|_2^2, \tag{1}$$

where $\mathbf{X}_{\cdot,-d}^{(\mathrm{obs})}$ denotes $\mathbf{X}^{(\mathrm{obs})}$ with the $d$-th column removed. This procedure is repeated for $d = 1, ..., \mathrm{D}$, yielding D univariate imputation models. In the inference stage, imputation is performed

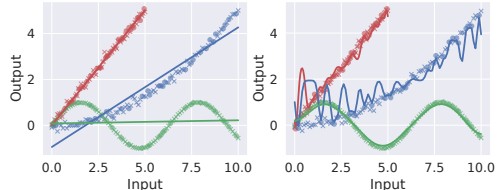
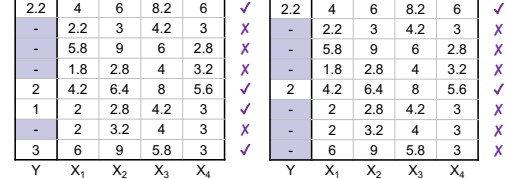

| 2.2 | 4   | 6   | 8.2 | 6   | ✓ |
|-----|-----|-----|-----|-----|---|
| -   | 2.2 | 3   | 4.2 | 3   | ✗ |
| -   | 5.8 | 9   | 6   | 2.8 | ✗ |
| -   | 1.8 | 2.8 | 4   | 3.2 | ✗ |
| 2   | 4.2 | 6.4 | 8   | 5.6 | ✓ |
| 1   | 2   | 2.8 | 4.2 | 3   | ✓ |
| -   | 2   | 3.2 | 4   | 3   | ✗ |
| 3   | 6   | 9   | 5.8 | 3   | ✓ |
| Y   | X₁  | X₂  | X₃  | X₄  |   |

| 2.2 | 4   | 6   | 8.2 | 6   | ✓ |
|-----|-----|-----|-----|-----|---|
| -   | 2.2 | 3   | 4.2 | 3   | ✗ |
| -   | 5.8 | 9   | 6   | 2.8 | ✗ |
| -   | 1.8 | 2.8 | 4   | 3.2 | ✗ |
| 2   | 4.2 | 6.4 | 8   | 5.6 | ✓ |
| -   | 2   | 2.8 | 4.2 | 3   | ✗ |
| -   | 2   | 3.2 | 4   | 3   | ✗ |
| -   | 6   | 9   | 5.8 | 3   | ✗ |
| Y   | X₁  | X₂  | X₃  | X₄  |   |

   (a) The implication of model misspecification.       (b) The implication of overlooking oracle features.

Figure 1: Case study illustrating the limitations of iterative imputation. In panel (a), circular and cross markers indicate observed and missing values, respectively, while lines represent imputation model outputs. In panel (b), "✓" denotes whether a sample can be used for training the imputation model for $Y$; dark areas indicate missing indices in $Y$ at missing ratios of 50% (left) and 75% (right).

iteratively: for each target feature, its missing values are estimated using the corresponding univariate model, conditioning on the current values of the remaining features (including values imputed in earlier steps). The resulting imputed features form the imputation matrix $\mathbf{X}^{(\mathrm{imp})}$.

# 3 Methodology

## 3.1 Motivation

In this section, we discuss two limitations inherent to standard iterative imputation. ❶ **It imposes a single parametric model form to impute all features**, such as a linear model [27] or a decision tree [30]. This rigidity risks model misspecification, as the optimal functional form often varies across features due to heterogeneous dependencies with other features [7]. For example, in a manufacturing process, temperature may depend linearly on pressure, while vibration could depend on pressure nonlinearly. A fixed parametric form cannot accommodate such heterogeneous dependencies, resulting in suboptimal imputation performance. ❷ **It assumes every feature contains missing values**. This assumption ignores the potential of oracle features—features that are fully or nearly non-missing—to provide high-quality supervision for the imputation process. In practice, oracle features are ubiquitous, such as demographic variables in healthcare records or reliable sensor readings in industrial systems. By neglecting these features, current iterative imputation suffers from limited supervisory signals, particularly in regimes with high missing ratios.

**Case study.** To substantiate the claims above, a case study is conducted. On limitation ❶, Fig. 1 (a) demonstrates that a fixed model form leads to model misspecification. In the left panel, a linear model accurately fits the linear feature well but underfits the others; in the right panel, a nonlinear model fits the sine feature well but overfits the others. Therefore, enforcing a single parametric model form risks model misspecification that limits imputation performance. On limitation ❷, Fig. 1 (b) illustrates that overlooking oracle features reduces the sample size usable for training $f_\theta$. To impute the target feature $Y$, iterative imputation fits $f_\theta$ using $X_1, \ldots, X_4$ as inputs, and trains it only on samples where $Y$ is observed. Under high missingness, only a handful of samples remain (two in the right panel), which is inadequate for learning a reliable estimation model. In contrast, $X_1, ..., X_4$ are non-missing oracle features, yet the method does not leverage them to provide additional supervision, thereby missing an opportunity to improve imputation accuracy.

These limitations underscore the need for an improved iterative imputation approach. In particular, there are three key questions that warrant investigation: (1) How to select the optimal model forms for each feature to reduce model misspecification? (2) How to incorporate oracle features for training imputation models? (3) Can these improvements enhance imputation performance in practice?

## 3.2 A bi-level optimization framework for iterative imputation

To address the two limitations above, we propose a novel bi-level optimization framework for iterative imputation. The framework learns feature-specific model forms within a reproducing kernel Hilbert space (RKHS) and integrates oracle features as explicit supervisory signals.

- To address limitation ❶, we replace the fixed parametric estimator ($f_\theta$ in (1)) with a nonparametric function $f$ optimized in an RKHS $\mathcal{H}$. Taking the $d$-th feature as the target feature, we reformulate the imputation task as a regularized functional optimization problem:

$$f^* = \arg\min_{f \in \mathcal{H}} \left\| \mathbf{Y}^{(\mathrm{obs})} - f(\mathbf{X}^{(\mathrm{obs})}_{\cdot,-d}) \right\|_2^2 + \lambda \|f\|_{\mathcal{H}}^2, \text{ with } \mathbf{Y}^{(\mathrm{obs})} = \mathbf{X}^{(\mathrm{obs})}_{\cdot,d}, \tag{2}$$

where $f^*$ is the optimal estimator for imputing the $d$-th feature. By optimizing over $\mathcal{H}$, the estimator can adapt its form to each feature, reducing the risk of misspecification.

- To address limitation ❷, we incorporate oracle features as supervisory signals for imputation. Suppose $\mathbf{X}^{(\mathrm{imp})} = \mathbf{X}^{(\mathrm{obs})} + \mathbf{X}^{(\mathrm{miss})}$, where $\mathbf{X}^{(\mathrm{miss})}$ contains the imputed values to be learned on missing entries ($\mathbf{M} = 1$) and is fixed to nan elsewhere. For an oracle feature $\mathbf{Y}^{(\mathrm{obs})}$ with its corresponding optimal estimator $f^*$, we refine the imputations of the remaining features by updating only their missing entries while keeping observed values fixed:

$$\min_{\mathbf{X}^{(\mathrm{miss})}} \left\| \mathbf{Y}^{(\mathrm{obs})} - f^* \left( \mathbf{X}^{(\mathrm{imp})}_{\cdot,-d} \right) \right\|_2^2. \tag{3}$$

This formulation is motivated by the following perspective: if $\mathbf{X}^{(\mathrm{imp})}$ is correctly imputed, the optimal estimator $f^*$ should produce estimates consistent with $\mathbf{Y}^{(\mathrm{obs})}$. Conversely, large estimation errors indicate that the current $\mathbf{X}^{(\mathrm{imp})}$ violates the dependency encoded by $f^*$. This process exploits oracle features as effective supervisory signals to update the other features.

Suppose $\mathbf{X} = \mathbf{X}^{(\mathrm{imp})}_{\cdot,-d}$ and $\mathbf{Y}^{(\mathrm{obs})} = \mathbf{X}^{(\mathrm{obs})}_{\cdot,d}$, combining (2) and (3) immediately yields:

$$\min_{\mathbf{X}^{(\mathrm{miss})}} \min_{f \in \mathcal{H}} \left\| \mathbf{Y}^{(\mathrm{obs})} - f(\mathbf{X}) \right\|_2^2 + \lambda \|f\|_{\mathcal{H}}^2, \tag{4}$$

which is performed for $d = 1, ..., \mathrm{D}$, treating each feature (including oracle features) in turn as the target feature $\mathbf{Y}^{(\mathrm{obs})}$. This bi-level optimization addresses both limitations of iterative methods.

### 3.3 Kernel function, universal property and learning objective

To solve the inner loop in (4), we approximate the optimum estimator $f^*$ via Gaussian kernels. We start by clarifying key properties of kernel functions in Definition 3.1 and 3.2.

**Definition 3.1** (Kernel function). Let $\mathcal{X}$ be a non-empty set. A function $K : \mathcal{X} \times \mathcal{X} \to \mathbb{R}$ is a kernel function if there exists a Hilbert space $\mathcal{H}$ and a feature map $\psi : \mathcal{X} \to \mathcal{H}$ such that $\forall x, x' \in \mathcal{X}$, $K(x, x') := \langle \psi(x), \psi(x') \rangle_{\mathcal{H}}$.

**Definition 3.2** (Universal kernel). For $\mathcal{X}$ a compact Hausdorff space, a universal kernel ensures that any continuous function $e : \mathcal{X} \to \mathbb{R}$ can be approximated arbitrarily well within the RKHS $\mathcal{H}$. Specifically, for any $\epsilon > 0$, there exists $f \in \mathcal{H}$ such that: $\sup_{x \in \mathcal{X}} |f(x) - e(x)| \le \epsilon$.

The Gaussian kernel is defined by $K(x, x') = \exp(-\|x - x'\|^2 / 2\sigma^2)$ and is universal in the sense of Definition 3.2 [29]. It implies that the RKHS of Gaussian kernel, defined as $\mathcal{H} = \mathrm{span}\{K(\cdot, x) \mid x \in \mathcal{X}\}$, admits uniform approximation of any continuous function.

**Lemma 3.3** (Representer theorem). *Suppose $h(\|f\|) : \mathbb{R}_+ \to \mathbb{R}$ is a non-decreasing function. The minimizer of an empirical risk functional regularized by $h(\|f\|)$ admits the form: $f^*(\cdot) = \sum_{i=1}^n \alpha_i K(\cdot, x_i)$ where $\boldsymbol{\alpha} = (\alpha_1, \ldots, \alpha_n)^\top$ and $K$ is the associated kernel function.*

**Lemma 3.4.** *Let $\mathbf{Y}^{\mathrm{s}}, \mathbf{Y}^{\mathrm{t}} \in \mathbb{R}^{\mathrm{B} \times 1}$ be the target feature and $\mathbf{X}^{\mathrm{s}}, \mathbf{X}^{\mathrm{t}}$ be the corresponding input features; Suppose $f^*$ is the optimal model minimizing the empirical risk in the inner optimization of (4), its output on $\mathbf{X}^{\mathrm{t}}$ is given by $f^*(\mathbf{X}^{\mathrm{t}}) = \mathbf{K}_{\mathbf{X}^{\mathrm{t}}\mathbf{X}^{\mathrm{s}}} \cdot \boldsymbol{\alpha}$, where $\boldsymbol{\alpha} = (\mathbf{K} + \lambda \mathbf{I})^{-1} \mathbf{Y}^{\mathrm{s}}$; $\mathbf{K}_{\mathbf{X}^{\mathrm{t}}\mathbf{X}^{\mathrm{s}}}$ is the kernel matrix computed with $\mathbf{X}^{\mathrm{t}}$ and $\mathbf{X}^{\mathrm{s}}$.*

Since $\mathcal{H}$ may be infinite-dimensional, directly identifying $f^* \in \mathcal{H}$ is intractable. Lemma 3.3 yields a finite-dimensional characterization of $f^*$, and Lemma 3.4 further implies that, given two batches of target features ($\mathbf{Y}^{\mathrm{s}}, \mathbf{Y}^{\mathrm{t}}$) and input features ($\mathbf{X}^{\mathrm{s}}, \mathbf{X}^{\mathrm{t}}$), the outputs of $f^*$ on $\mathbf{X}^{\mathrm{t}}$ admit the closed form:

$$f^*(\mathbf{X}^{\mathrm{t}}) = \mathbf{K}_{\mathbf{X}^{\mathrm{t}}\mathbf{X}^{\mathrm{s}}}(\mathbf{K}_{\mathbf{X}^{\mathrm{s}}\mathbf{X}^{\mathrm{s}}} + \lambda \mathbf{I})^{-1} \mathbf{Y}^{\mathrm{s}}, \tag{5}$$

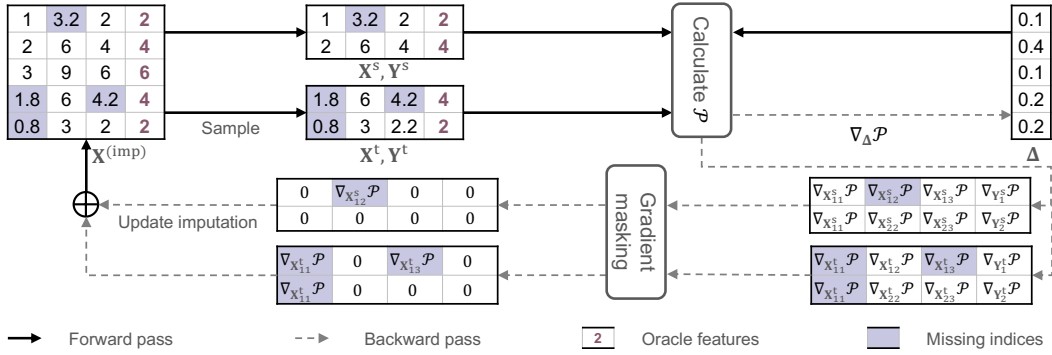

Figure 2: Visualization of the workflow of KPI, where the dataset contains 5 samples and 4 features. The sampling batch size is set to 2. The last column is the oracle feature without missing values.

Therefore, the output of $f^*$ is expressed as a combination of kernel evaluations, retaining feature-wise adaptation of model form while reducing (4) to a differentiable loss function:

$$\min_{\mathbf{X}^s, \mathbf{X}^t} \left\| \mathbf{Y}^t - \mathbf{K}_{\mathbf{X}^t \mathbf{X}^s} (\mathbf{K}_{\mathbf{X}^s \mathbf{X}^s} + \lambda \mathbf{I})^{-1} \mathbf{Y}^s \right\|_2^2. \tag{6}$$

The tuning of kernel hyperparameters (e.g., the Gaussian bandwidth) is notably challenging amidst incomplete data. To alleviate this problem, we employ multiple kernels with distinct parameters and learn to ensemble them adaptively. Suppose $\mathbf{K}^1, \mathbf{K}^2, ..., \mathbf{K}^E$ are kernel matrices with different hyperparameters, $\mathbf{\Delta} \in \mathbb{R}^K$ is a learnable simplex vector; we form the ensemble kernel as $\mathbf{K}^{\mathbf{\Delta}} = \mathbf{K}^1 \mathbf{\Delta}_1 + ... + \mathbf{K}^E \mathbf{\Delta}_E$. Putting together, the final objective becomes:

$$\mathcal{P} = \left\| \mathbf{Y}^t - \mathbf{K}^{\mathbf{\Delta}}_{\mathbf{X}^t \mathbf{X}^s} (\mathbf{K}^{\mathbf{\Delta}}_{\mathbf{X}^s \mathbf{X}^s} + \lambda \mathbf{I})^{-1} \mathbf{Y}^s) \right\|_2^2. \tag{7}$$

### 3.4 Overall workflow

While the learning objective is well defined, how to use it for imputation remains unclear. To this end, we propose the kernel point imputation (KPI) method, which iteratively minimizes the objective (7) to refine imputation of missing values. The core procedure is shown in Fig. 2 and detailed as follows.

**Initialization.** Given the incomplete dataset $\mathbf{X}^{(obs)}$, we initialize missing values using the mean of observed steps, obtaining an initial imputation matrix $\mathbf{X}^{imp}$. The imputed values are treated as learnable parameters, and their gradients are tracked throughout training.

**Forward Pass.** Two batches are sampled from the imputation matrix. In each iteration, a column is randomly chosen as the target feature ($\mathbf{Y}^s, \mathbf{Y}^t \in \mathbb{R}^{B \times 1}$), with the remaining columns as input features ($\mathbf{X}^s, \mathbf{X}^t \in \mathbb{R}^{B \times (D-1)}$), where B represents batch size, s and t differentiates different batches. The objective $\mathcal{P}$ is computed following (7).

**Backward Pass.** The gradients of $\mathcal{P}$ with respect to $\mathbf{X}^s$, $\mathbf{X}^t$ and $\mathbf{\Delta}$ are calculated using automatic differentiation. The imputed values in $\mathbf{X}^s$ and $\mathbf{X}^t$ as well as $\mathbf{\Delta}$ are then updated using gradient descent with an update rate $\eta$:

$$\begin{aligned}
\mathbf{X}^s &\leftarrow \mathbf{X}^s - \eta \nabla_{\mathbf{X}^s} \mathcal{P} \odot \mathbf{M}^s, \\
\mathbf{X}^t &\leftarrow \mathbf{X}^t - \eta \nabla_{\mathbf{X}^t} \mathcal{P} \odot \mathbf{M}^t, \\
\mathbf{\Delta} &\leftarrow \mathbf{\Delta} - \eta \nabla_{\mathbf{\Delta}} \mathcal{P},
\end{aligned} \tag{8}$$

where only the missing values (with $\mathbf{M} = 1$) are updated, while the observed values (with $\mathbf{M} = 0$) remain unchanged during this process. Moreover, the gradient of the matrix inverse term is stopped for numerical stability. KPI iteratively executes the forward and backward passes sampling different batches until hitting the early-stopping criteria on the validation dataset. In this process, each feature is iteratively treated as the target feature while the remaining features are treated as the input features. This ensures that all features—including oracle features—are fully exploited as supervisory signals.

Table 1: Imputation performance in terms of MSE and MAE.

| Datasets | BT | | CC | | CBV | | IS | | PK | | QB | | WQW | |
|---|---|---|---|---|---|---|---|---|---|---|---|---|---|---|
| Metrics | MSE | MAE | MSE | MAE | MSE | MAE | MSE | MAE | MSE | MAE | MSE | MAE | MSE | MAE |
| Mean | 0.742 | 0.452 | 0.837 | 0.789 | 0.829 | 1.165 | 0.754 | 4.145 | 0.740 | 2.841 | 0.589 | 4.682 | 0.764 | 1.121 |
| Mode | 0.948 | 0.770 | 0.935 | 1.159 | 1.026 | 1.749 | 0.925 | 7.741 | 1.254 | 7.832 | 0.593 | 6.240 | 0.823 | 1.372 |
| Median | 0.706 | 0.469 | 0.811 | 0.884 | 0.820 | 1.165 | 0.713 | 4.356 | 0.698 | 3.029 | 0.500 | 5.066 | 0.756 | 1.123 |
| MICE | 0.580 | 0.127 | 0.745 | 0.474 | 0.856 | 1.021 | 0.733 | 4.539 | 0.417 | 1.312 | 0.536 | 3.415 | 0.824 | 0.971 |
| Miss.F | 0.560 | 0.241 | 0.732 | 0.650 | 0.764 | 0.994 | 0.593 | 3.277 | 0.526 | 1.497 | 0.436 | 3.202 | 0.686 | 0.898 |
| Sinkhorn | 0.835 | 0.466 | 0.906 | 0.796 | 0.898 | 1.225 | 0.848 | 4.945 | 0.827 | 3.233 | 0.775 | 6.114 | 0.857 | 1.170 |
| TDM | 0.730 | 0.487 | 0.819 | 0.769 | 0.799 | 1.113 | 0.726 | 3.965 | 0.722 | 2.792 | 0.570 | 4.756 | 0.752 | 1.098 |
| CSDI-T | 0.726 | 1.870 | 0.849 | 2.683 | 0.821 | 3.802 | 0.761 | 15.493 | 0.731 | 12.291 | 0.575 | 19.919 | 0.780 | 4.084 |
| MissDiff | 0.719 | 1.332 | 0.840 | 1.699 | 0.816 | 3.523 | 0.749 | 13.432 | 0.728 | 14.462 | 0.564 | 23.320 | 0.758 | 5.184 |
| GAIN | 0.730 | 0.396 | 0.777 | 0.688 | 0.729 | 0.942 | 0.572 | 3.318 | 0.448 | 1.413 | 0.476 | 4.669 | 0.754 | 1.095 |
| MIRACLE | 0.795 | 0.674 | 0.487 | 0.305 | 0.831 | 1.154 | 3.208 | 45.816 | 3.518 | 36.784 | 0.521 | 3.975 | 0.555 | 0.685 |
| MIWAE | 0.582 | 0.266 | 0.746 | 0.630 | 0.807 | 1.071 | 0.636 | 4.118 | 0.525 | 1.804 | 0.475 | 4.977 | 0.657 | 0.844 |
| Remasker | 0.439 | 0.131 | 0.767 | 0.750 | 0.528 | 0.522 | 0.599 | 3.584 | 0.447 | 1.268 | 0.401 | 2.811 | 0.546 | 0.636 |
| NewImp | 0.465 | 0.177 | 0.412 | 0.292 | 0.405 | 0.401 | 0.431 | 2.495 | 0.320 | 0.857 | 0.332 | 2.992 | 0.497 | 0.692 |
| **KPI(Ours)** | **0.397** | **0.121** | **0.347** | **0.284** | **0.402** | **0.394** | **0.400** | **2.387** | **0.319** | **0.747** | **0.264** | **2.131** | **0.491** | **0.685** |

*Note*: Each entry represents the average results at four missing ratios: 0.1, 0.2, 0.3, and 0.4. The best and second-best results are **bolded** and underlined, respectively.

## 4 Experiments

### 4.1 Experimental setup

- **Datasets:** The empirical study is performed on public tabular datasets from [1], including Blood Transfusion(BT), Concrete Compression (CC), Connectionist Bench Vowel (CBV), Ionosphere (IS) Parkinsons (PS), Qsar Biodegradation (QB), and Wine Quality White (QWQ). We simulate missing data via a binary mask matrix, characterized by a predefined missing ratio. The specific generation strategies of mask matrix in different missing scenarios are detailed in Appendix B.1.

- **Baselines:** The performance of KPI is compared against various imputation methods, including iterative imputers (MICE [27], Miss.F. [30]), and generative models (GAIN [42], MIWAE [20], Miss.D [25], CSDI-T [32], ReMasker [4], and NewImp [1]). We also assess methods that do not conform to these categories, such as MIRACLE [9], Sinkhorn [24], and TDM [46].

- **Implementation details:** To ensure convergence, we cap the number of iterations at 500 and adopt an early stopping criterion based on validation performance, with a patience of 10 epochs. The Adam optimizer is used for training [8]. Key hyperparameters, namely $\eta$ and B, are determined by allocating 5% of the training data for validation and finetuning over $[0.0001, 0.01]$ for $\eta$ and $[64, 512]$ for B. Performance is assessed using modified mean absolute error (MAE) and mean squared error (MSE), focusing on the imputed values at missing values, following [46, 7]. In addition, we report the distribution discrepancy (WASS), measured as the Wasserstein distance [7]. The experiments are performed on a platform with two Intel(R) Xeon(R) Platinum 8383C CPUs @ 2.70GHz and a NVIDIA GeForce RTX 4090 GPU.

### 4.2 Overall performance

Tab. 1 presents the average imputation results of KPI and baseline methods under missing ratios $p_{\text{miss}} = 0.1, 0.2, 0.3,$ and $0.4$. Key observations are summarized as follows:

- The iterative imputers exhibits promising performance in most cases. For instance, MICE outperforms simple imputers by large margin over most datasets. MissForest employs random forest as the base model, excelling in handling tabular data, which further improves imputation quality.

- The canonical generative imputers [32, 25], originally tailored for time-series data, often falling behind iterative methods. This can be attributed to the implicit maximization of imputation

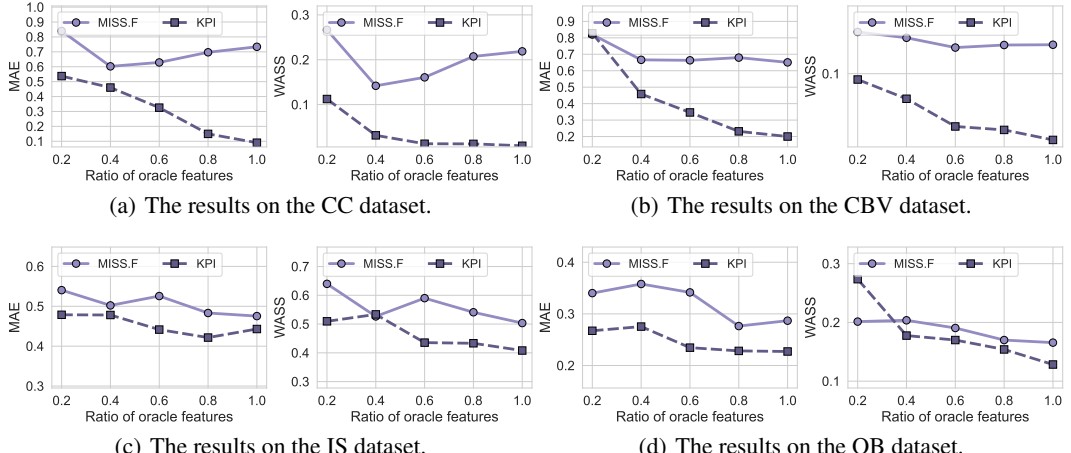

(a) The results on the CC dataset.    (b) The results on the CBV dataset.

(c) The results on the IS dataset.    (d) The results on the QB dataset.

Figure 3: The performance of Miss.F and KPI given varying ratios of oracle features.

Table 2: Varying kernel number results.

| | CC | | | | | |
|---|---|---|---|---|---|---|
| E | MSE | ΔMSE | WASS | ΔWASS | MAE | ΔMAE |
| 1 | 0.082 | - | 0.058 | - | 0.155 | - |
| 3 | 0.070 | 14.6%↓ | 0.055 | 5.2%↓ | 0.116 | 25.2%↓ |
| 5 | 0.069 | 15.9%↓ | 0.046 | 20.7%↓ | 0.108 | 30.3%↓ |
| 7 | 0.065 | 20.7%↓ | 0.039 | 32.8%↓ | 0.091 | 41.3%↓ |
| | CBV | | | | | |
| E | MSE | ΔMSE | WASS | ΔWASS | MAE | ΔMAE |
| 1 | 0.128 | - | 0.095 | - | 0.233 | - |
| 3 | 0.110 | 14.1%↓ | 0.085 | 10.5%↓ | 0.226 | 3.0%↓ |
| 5 | 0.098 | 23.4%↓ | 0.075 | 21.1%↓ | 0.216 | 7.3%↓ |
| 7 | 0.087 | 32.0%↓ | 0.066 | 30.5%↓ | 0.205 | 12.0%↓ |
| | BT | | | | | |
| E | MSE | ΔMSE | WASS | ΔWASS | MAE | ΔMAE |
| 1 | 0.334 | - | 0.109 | - | 0.363 | - |
| 3 | 0.318 | 4.8%↓ | 0.101 | 7.3%↓ | 0.352 | 3.0%↓ |
| 5 | 0.305 | 8.7%↓ | 0.096 | 11.9%↓ | 0.343 | 5.5%↓ |
| 7 | 0.302 | 9.6%↓ | 0.089 | 18.3%↓ | 0.338 | 6.9%↓ |

Table 3: Varying kernel function results.

| | CC | | | | | |
|---|---|---|---|---|---|---|
| Kernel | MSE | ΔMSE | WASS | ΔWASS | MAE | ΔMAE |
| Linear | 0.099 | - | 0.051 | - | 0.203 | - |
| Poly | 0.065 | 34.3%↓ | 0.039 | 23.5%↓ | 0.091 | 55.2%↓ |
| Laplacian | 0.068 | 31.3%↓ | 0.042 | 17.6%↓ | 0.093 | 54.2%↓ |
| Gaussian | 0.076 | 23.2%↓ | 0.035 | 31.4%↓ | 0.082 | 59.6%↓ |
| | CBV | | | | | |
| Kernel | MSE | ΔMSE | WASS | ΔWASS | MAE | ΔMAE |
| Linear | 0.089 | - | 0.087 | - | 0.219 | - |
| Poly | 0.087 | 2.2%↓ | 0.088 | 1.1% ↑ | 0.214 | 2.3%↓ |
| Laplacian | 0.083 | 6.7%↓ | 0.080 | 8.1%↓ | 0.211 | 3.7%↓ |
| Gaussian | 0.087 | 2.2%↓ | 0.066 | 24.1%↓ | 0.205 | 6.4%↓ |
| | BT | | | | | |
| Kernel | MSE | ΔMSE | WASS | ΔWASS | MAE | ΔMAE |
| Linear | 0.326 | - | 0.101 | - | 0.359 | - |
| Poly | 0.305 | 6.4%↓ | 0.090 | 10.9%↓ | 0.342 | 4.7%↓ |
| Laplacian | 0.316 | 3.1%↓ | 0.091 | 9.9%↓ | 0.346 | 3.6%↓ |
| Gaussian | 0.302 | 7.4%↓ | 0.089 | 11.9%↓ | 0.338 | 5.8%↓ |

entropy in diffusion models, which negatively impacts accuracy [1]. By contrast, recent generative approaches such as NewImp and Remasker handle this issue and achieve strong results, obtaining the best results among baseline methods.

- KPI improves the iterative imputers by adaptively selecting the optimal imputer for each feature and involving oracle features as supervisory signals. This strategy consistently improves performance, as evidenced by KPI outperforming all baselines across all 7 datasets—often by a substantial margin, particularly on the CC and QB datasets—demonstrating strong practical effectiveness.

## 4.3 Impact of oracle features

In this section, we assess the impact of using oracle features as supervisory signal on imputation performance. Specifically, we simulate based on complete datasets to generate varying ratios of oracle features and evaluate the imputation performance. Two models are considered: KPI and another canonical iterative imputer: Miss.F.

The results are presented in Fig. 3. As the ratio of oracle features increases, KPI consistently exhibits lower imputation error, showcasing the utility of oracle features. In contrast, Miss.F shows little improvement as oracle feature ratio increases. This difference arises because Miss.F only uses oracle features as inputs, whereas KPI exploits them as supervisory signals to refine the imputation results.

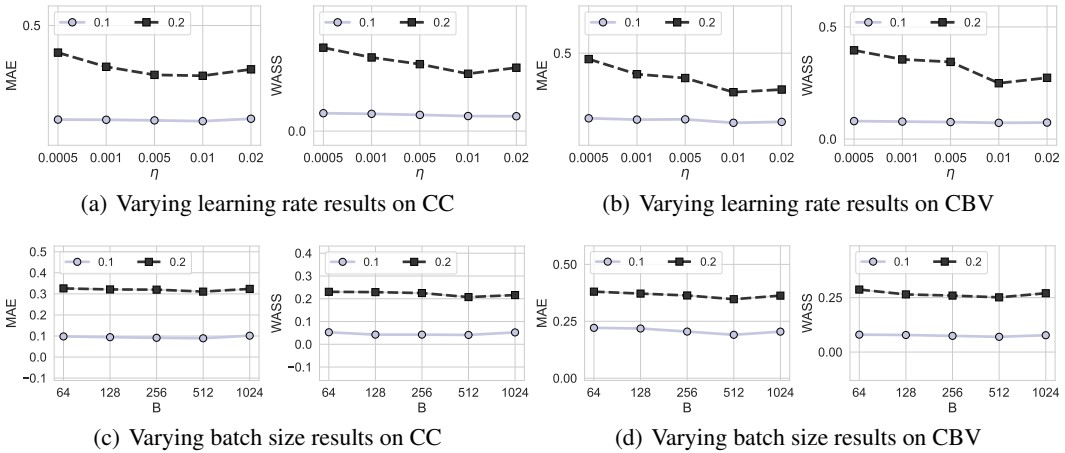

(a) Varying learning rate results on CC

(b) Varying learning rate results on CBV

(c) Varying batch size results on CC

(d) Varying batch size results on CBV

Figure 4: Varying learning rate and batch size results with missing ratios 0.1 and 0.2.

## 4.4 Impact of kernel strategy

In this section, we analyze the impact of kernel function and kernel amount (E) on imputation performance. The key observations are summarized as follows.

- The multiple kernel ensembling mechanism has a substantial impact. As shown in Tab. 2, increasing E from 1 to 7 consistently reduces MSE from 0.082 to 0.065, indicating a relative reduction of 20.7%. This gain is attributed to the increased flexibility in adaptively selecting kernel parameters, allowing KPI to better represent the optimal imputation model for each feature.

- The performance of different kernel functions showcases the importance of kernel universality. The linear kernel, which is not universal and has limited RKHS capacity, yields the worst performance. The polynomial kernel, with a larger RKHS, performs better. The Gaussian kernel exhibits the best overall performance. The superiority is attributed to its universality, i.e., the associated RKHS admits uniform approximation of any continuous function. Such extensive RKHS capacity enables KPI to optimize the imputation model for each feature, thereby enhancing imputation performance.

## 4.5 Parameter sensitivity analysis

In this section, we examine the influence of critical hyperparameters on the performance of KPI in Fig. 4. Below are the key observations:

- The update rate ($\eta$) plays a pivotal role in controlling the volume of updates to the imputation matrix each epoch. As $\eta$ is reduced from 0.02 to approximately 0.01, both MAE and RMSE decrease, indicating that a smaller $\eta$ enhances update stability. However, further reduction of $\eta$ to 0.001 results in increased MAE and RMSE, where the meaningful update direction becomes overshadowed by noise, preventing model convergence within the allocated epochs.

- The batch size (B) affects the scale of the problem in calculating discrepancies, with sizes ranging from 64 to 1024 examined. There is a weak yet consistent decrease in MAE and RMSE as the batch size increases to $B = 512$, enhancing the reliability of estimations. Increasing the batch size beyond this point yields diminishing returns and may lead to unnecessary computational overhead.

## 5 Related works

The pervasive presence of missing data undermines the integrity of collected datasets and the reliability of data-driven applications, underscoring the necessity for effective missing data imputation (MDI). To achieve accurate MDI, existing approaches can be broadly categorized into two paradigms: discriminative and generative, each with distinct advantages and limitations [7, 22].

The iterative method [30, 27, 44, 16] is one of the most popular methods in discriminative imputation, initiated from imputation by chained equations (ICE) [27], which employs specific models to estimate

missing values for each feature based on the remaining observable features. On the basis of ICE, a line of work advocates for employing modern parametric models, such as neural networks [20, 9], Bayesian models [27] and random forest [30], which enhances the capacity of imputation models and thereby accommodating complex missing patterns. In a different line of work, various training techniques are investigated within the paradigm, such as multiple imputation [27], ensemble learning [30], and multitask learning [20], which enhance its utility in diverse contexts. While this paradigm offers enhanced flexibility and accuracy, it fails to utilize the oracle features effectively and risks model misspecification, which can lead to suboptimal imputation results in noisy environments. Our research advances this methodology by handling the two limitations.

Apart from the iterative methods, there are other notable approaches in the discriminative paradigm. The simple direct paradigm employs elementary statistical measures like mean, median, and mode to replace missing values, offering quick and straightforward solutions. However, this approach lacks the capacity to accommodate complex dependencies [18, 21], often producing trivial and inadequate imputation results that fail to meet expectations in practice. Another notable approach is matrix factorization, which decomposes the data matrix into two low-rank matrices, capturing the latent structure of the data for imputation [10, 6]. This method is particularly effective in collaborative filtering and recommendation systems [14, 37]. Recent advances explore a novel methodology based on distribution discrepancy minimization[46, 24]. This approach builds on the assumption that, under the independent and identically distributed (i.i.d.) condition, any two data batches should share the same underlying distribution, thereby exhibiting minimal discrepancy. Subsequent studies have extended this idea by refining discrepancy measures to accommodate different data characteristics such as neighboring effects [39, 35], noisy observations [36], and temporal dependencies [38].

The generative paradigm restates imputation as a conditional generation problem, using advanced generative models, such as generative adversarial networks [42, 31, 15] and diffusions [32, 41, 1], to approximate data distributions and perform imputation. This strategy incorporates the strengths of generative models, capturing and utilizing complex dependencies, which potentially enhances the imputation quality when ample data is available. However, it also bears the defects with generative models, such as the instability associated with adversarial training and the operational complexity of diffusions [19, 26], hampering their use in practice.

# 6  Conclusion

Iterative imputation methods are widely used for handling missing data, yet existing approaches are often limited by model misspecification and underuse of oracle features. To overcome these limitations, we introduce KPI, a bi-level optimization framework which optimizes model form within RKHS for each feature, reducing model misspecification, and exploits oracle features as effective supervision. Extensive experiments on real-world datasets demonstrate that KPI achieves superior imputation performance and effectively leverages oracle features.

**Limitations & future work.** In this work, we do not accommodate potential noise in datasets, which is a prevalent challenge in industrial settings [6, 5]. Future research could incorporate robust optimization techniques and truncate outliers in the kernel matrix which has potential to improve noise robustness. Additionally, this work mitigates the difficulty of concise kernel parameter selection via adaptive ensembling, which is an heuristic approach. Subsequent work may explore meta-learning strategies with theoretical guarantees for accurate kernel parameter selection.

# Acknowledgments

Z. Lin was supported by the NSF China (No. 62276004) and the State Key Laboratory of General Artificial Intelligence. H. Li was supported by National Natural Science Foundation of China (623B2002).

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

# A   Theoretical justification

In this section, we delve into the theoretical underpinnings of the KPI framework. By exploiting the universality, injectivity, and reproducing properties of kernel functions, we derive formal guarantees for KPI and demonstrate how these properties translate into effective imputation.

**Lemma A.1** (Representer theorem). *Suppose $h(\|f\|) : \mathbb{R}_+ \to \mathbb{R}$ is a non-decreasing function. The minimizer of an empirical risk functional regularized by $h(\|f\|)$ admits the form: $f^*(\cdot) = \sum_{i=1}^n \alpha_i K(\cdot, x_i)$ where $\boldsymbol{\alpha} = (\alpha_1, \ldots, \alpha_n)^\top$ and $K$ is the associated kernel function.*

*Proof.* The proof can be found in Theorem 6.11 of Mohri et al. [23]. $\qquad\square$

**Theorem A.2** (Lemma 3.4 in the main text). *Let $\mathbf{Y}^s, \mathbf{Y}^t \in \mathbb{R}^{B \times 1}$ be the target feature and $\mathbf{X}^s$, $\mathbf{X}^t$ be the corresponding input features; Suppose $f^*$ is the optimal model minimizing the empirical risk in the inner optimization of* (4)*, its output on $\mathbf{X}^t$ is given by $f^*(\mathbf{X}^t) = \mathbf{K}_{\mathbf{X}^t \mathbf{X}^s} \cdot \boldsymbol{\alpha}$, where $\boldsymbol{\alpha} = (\mathbf{K} + \lambda \mathbf{I})^{-1} \mathbf{y}$; $\mathbf{K}_{\mathbf{X}^t \mathbf{X}^s}$ is the kernel matrix computed with $\mathbf{X}^t$ and $\mathbf{X}^s$.*

*Proof.* This is a standard theoretical result in kernel ridge regression, and we detail its proof here with contextualization. Consider the samples $\mathbf{X}^s$ and $\mathbf{Y}^s$ where $\mathbf{Y}^s$ is the observed target, and $\mathbf{X}^s$ comprises the input features. The empirical risk minimization objective with $\ell_2$ regularization to select the optimal functional form is

$$\min_{f \in \mathcal{H}} \|\mathbf{Y}^s - f(\mathbf{X}^s)\|_2^2 + \lambda \|f\|_{\mathcal{H}}^2, \tag{9}$$

which corresponds precisely to the inner loop of (4). According to Lemma A.1, when $h$ is an identity function (in (9)) and $\mathcal{H}$ is a RKHS associated with kernel $K$, the minimizer $f^*$ must admit the explicit form

$$f^*(x) = \sum_{i=1}^{B} \alpha_i K(x, x_i^s), \tag{10}$$

for some coefficients $\alpha_1, \ldots, \alpha_B$.

Substituting this form into the empirical risk (9), the optimization problem becomes

$$\min_{\boldsymbol{\alpha} \in \mathbb{R}^B} \|\mathbf{Y}^s - \mathbf{K}_{\mathbf{X}^s \mathbf{X}^s} \boldsymbol{\alpha}\|_2^2 + \lambda \boldsymbol{\alpha}^\top \mathbf{K}_{\mathbf{X}^s \mathbf{X}^s} \boldsymbol{\alpha}, \tag{11}$$

where $\mathbf{K}_{\mathbf{X}^s \mathbf{X}^s}$ is the $B \times B$ Gram matrix, with $(i, j)$-th entry $K(x_i^s, x_j^s)$, $\mathbf{Y}^s$ is the length-B target vector, and $\boldsymbol{\alpha}$ is the vector of coefficients.

Expanding the loss function in matrix notation yields

$$\left(\mathbf{Y}^s - \mathbf{K}_{\mathbf{X}^s \mathbf{X}^s} \boldsymbol{\alpha}\right)^\top \left(\mathbf{Y}^s - \mathbf{K}_{\mathbf{X}^s \mathbf{X}^s} \boldsymbol{\alpha}\right) + \lambda \boldsymbol{\alpha}^\top \mathbf{K}_{\mathbf{X}^s \mathbf{X}^s} \boldsymbol{\alpha}. \tag{12}$$

Due to symmetry of $\mathbf{K}_{\mathbf{X}^s \mathbf{X}^s}$, this simplifies to

$$\mathbf{Y}^{s\top} \mathbf{Y}^s - 2 \mathbf{Y}^{s\top} \mathbf{K}_{\mathbf{X}^s \mathbf{X}^s} \boldsymbol{\alpha} + \boldsymbol{\alpha}^\top (\mathbf{K}_{\mathbf{X}^s \mathbf{X}^s}^2 + \lambda \mathbf{K}_{\mathbf{X}^s \mathbf{X}^s}) \boldsymbol{\alpha}. \tag{13}$$

According to the first-order condition, setting the derivative with respect to $\boldsymbol{\alpha}$ to zero and solving for $\boldsymbol{\alpha}$ gives

$$-2 \mathbf{K}_{\mathbf{X}^s \mathbf{X}^s}^\top \mathbf{Y}^s + 2 (\mathbf{K}_{\mathbf{X}^s \mathbf{X}^s}^2 + \lambda \mathbf{K}_{\mathbf{X}^s \mathbf{X}^s}) \boldsymbol{\alpha} = 0, \tag{14}$$

which is equivalent to:

$$\mathbf{K}_{\mathbf{X}^s \mathbf{X}^s} (\mathbf{K}_{\mathbf{X}^s \mathbf{X}^s} + \lambda \mathbf{I}) \boldsymbol{\alpha} = \mathbf{K}_{\mathbf{X}^s \mathbf{X}^s} \mathbf{Y}^s. \tag{15}$$

Assuming $\mathbf{K}_{\mathbf{X}^s \mathbf{X}^s}$ is invertible, we have:

$$(\mathbf{K}_{\mathbf{X}^s \mathbf{X}^s} + \lambda \mathbf{I}) \boldsymbol{\alpha} = \mathbf{Y}^s. \tag{16}$$

which immediately follows from multiplying both sides by $\mathbf{K}_{\mathbf{X}^s \mathbf{X}^s}^{-1}$. Solving for $\boldsymbol{\alpha}$ gives:

$$\boldsymbol{\alpha} = (\mathbf{K}_{\mathbf{X}^s \mathbf{X}^s} + \lambda \mathbf{I})^{-1} \mathbf{Y}^s. \tag{17}$$

Substituting (17) into (10) leads to

$$f^*(x) = \sum_{i=1}^{B} \alpha_i K(x, x_i^s) = \mathbf{K}(x)(\mathbf{K}_{\mathbf{X}^s \mathbf{X}^s} + \lambda \mathbf{I})^{-1} \mathbf{Y}^s, \tag{18}$$

where $\mathbf{K}(x)$ is the $1 \times \mathrm{B}$ vector $[K(x, x_1^{\mathrm{s}}), \cdots, K(x, x_{\mathrm{B}}^{\mathrm{s}})]$. For a (possibly distinct) batch of inputs $\mathbf{X}^{\mathrm{t}}$, evaluating $f^*$ at each $x_j^{\mathrm{t}}$ gives

$$f^*(x_1^{\mathrm{t}}) = \sum_{i=1}^{\mathrm{B}} \alpha_i K(x_1^{\mathrm{t}}, x_i^{\mathrm{s}}) = \left[K(x_1^{\mathrm{t}}, x_1^{\mathrm{s}}), K(x_1^{\mathrm{t}}, x_2^{\mathrm{s}}), ..., K(x_1^{\mathrm{t}}, x_{\mathrm{B}}^{\mathrm{s}})\right] (\mathbf{K}_{\mathbf{X}^{\mathrm{s}}\mathbf{X}^{\mathrm{s}}} + \lambda\mathbf{I})^{-1}\mathbf{Y}^{\mathrm{s}},$$

$$f^*(x_2^{\mathrm{t}}) = \sum_{i=1}^{\mathrm{B}} \alpha_i K(x_2^{\mathrm{t}}, x_i^{\mathrm{s}}) = \left[K(x_2^{\mathrm{t}}, x_1^{\mathrm{s}}), K(x_2^{\mathrm{t}}, x_2^{\mathrm{s}}), ..., K(x_2^{\mathrm{t}}, x_{\mathrm{B}}^{\mathrm{s}})\right] (\mathbf{K}_{\mathbf{X}^{\mathrm{s}}\mathbf{X}^{\mathrm{s}}} + \lambda\mathbf{I})^{-1}\mathbf{Y}^{\mathrm{s}},$$

$$...$$

$$f^*(x_{\mathrm{B}}^{\mathrm{t}}) = \sum_{i=1}^{\mathrm{B}} \alpha_i K(x_{\mathrm{B}}^{\mathrm{t}}, x_i^{\mathrm{s}}) = \left[K(x_{\mathrm{B}}^{\mathrm{t}}, x_1^{\mathrm{s}}), K(x_{\mathrm{B}}^{\mathrm{t}}, x_2^{\mathrm{s}}), ..., K(x_{\mathrm{B}}^{\mathrm{t}}, x_{\mathrm{B}}^{\mathrm{s}})\right] (\mathbf{K}_{\mathbf{X}^{\mathrm{s}}\mathbf{X}^{\mathrm{s}}} + \lambda\mathbf{I})^{-1}\mathbf{Y}^{\mathrm{s}},$$

$$(19)$$

which may be stacked to give the vector-valued expression

$$f^*(\mathbf{X}^{\mathrm{t}}) = \mathbf{K}_{\mathbf{X}^{\mathrm{t}}\mathbf{X}^{\mathrm{s}}}(\mathbf{K}_{\mathbf{X}^{\mathrm{s}}\mathbf{X}^{\mathrm{s}}} + \lambda\mathbf{I})^{-1}\mathbf{Y}^{\mathrm{s}}, \tag{20}$$

where $\mathbf{K}_{\mathbf{X}^{\mathrm{t}}\mathbf{X}^{\mathrm{s}}}$ is the matrix with entries $[\mathbf{K}_{\mathbf{X}^{\mathrm{t}}\mathbf{X}^{\mathrm{s}}}]_{ij} = K(x_i^{\mathrm{t}}, x_j^{\mathrm{s}})$. The proof is completed. $\qquad\square$

**Definition A.3** (Kernel Functions). Let $\mathbf{x}, \mathbf{x}' \in \mathbb{R}^{\mathrm{D}}$ be two vectors in the input feature space. A *kernel function* $K : \mathbb{R}^{\mathrm{D}} \times \mathbb{R}^{\mathrm{D}} \to \mathbb{R}$ is a symmetric, positive semi-definite function that quantifies the similarity between $\mathbf{x}$ and $\mathbf{x}'$. Commonly used kernel functions include:

1. **Linear Kernel:** $K_{\mathrm{linear}}(\mathbf{x}, \mathbf{x}') = \mathbf{x}^\top \mathbf{x}'$, which computes the inner product between two vectors and corresponds to the case where no explicit feature transformation is applied.

2. **Polynomial Kernel:** $K_{\mathrm{poly}}(\mathbf{x}, \mathbf{x}') = \left(\mathbf{x}^\top \mathbf{x}' + c\right)^d$, where $c \geq 0$ is a constant coefficient trading off the influence of higher-order versus lower-order terms, and $d \in \mathbb{N}$ is the degree of the polynomial. It enables learning non-linear dependencies by implicitly mapping the input features into a higher-dimensional polynomial feature space.

3. **Gaussian Kernel:** $K_{\mathrm{gauss}}(\mathbf{x}, \mathbf{x}') = \exp\left(-\frac{\|\mathbf{x}-\mathbf{x}'\|^2}{2\sigma^2}\right)$, where $\|\mathbf{x} - \mathbf{x}'\|^2$ denotes the squared Euclidean distance between $\mathbf{x}$ and $\mathbf{x}'$, and $\sigma > 0$ is a scale parameter controlling the width of the kernel. The Gaussian kernel is widely used due to its ability to model localized and highly non-linear interactions.

# B  Implementation details

## B.1  Dataset description and process strategy

In this paper, we use UCI datasets for model validation. To simulate missing data, we first construct a binary mask matrix $\mathbf{M}$. The observed data matrix, denoted as $\mathbf{X}^{(\mathrm{obs})}$, is derived by element-wise application of the complement mask $\mathbf{1} - \mathbf{M}$ to the non-missing data matrix $\mathbf{X}^{(\mathrm{id})}$. On the generation of $\mathbf{M}$, we consider three canonical missing data mechanisms:

- **Missing Completely at Random (MCAR):** The probability of entry-wise missingness is independent of both observed and unobserved data. To simulate MCAR, each entry of $\mathbf{M}$ is independently set to 1 (missing) with probability $p_{\mathrm{miss}}$, and to 0 (observed) with probability $1 - p_{\mathrm{miss}}$.

- **Missing at Random (MAR):** The missingness of a variable depends only on values of observed variables [34, 11]. To generate MAR scenarios, we randomly select a subset of features to be always observed. The missingness in the remaining features is simulated using a logistic regression model, where the observed features act as inputs. The model parameters are randomly initialized, and the bias is calibrated to yield the desired missing ratio.

- **Missing Not at Random (MNAR):** The probability that a value is missing depends on the values themselves [37, 13]. For MNAR simulation, we adopt the procedure in [1, 7]: the logistic model used for MAR is repurposed, but its inputs are themselves masked by an independent MCAR mechanism, making the missingness dependent on both observed and unobserved features.

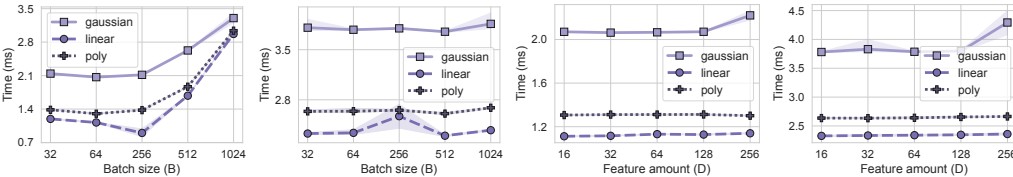

(a) Computation time with different batch sizes.  (b) Computation time with different feature amounts.

Figure 5: Running time of the forward pass (left panels) and backward pass (right panels) given varying settings. Different colors indicate different kernel functions. The colored lines and the shadowed areas indicate the mean values and the 99.9% confidence intervals.

### B.2 Training protocols

To ensure reliable convergence, we set a maximum of 500 training iterations and adopt early stopping based on validation performance, using a patience parameter of 10 epochs. Optimization throughout is conducted using the Adam optimizer [8]. The kernel function is specified as the Gaussian kernel. The main hyperparameters, specifically the update rate $\eta$, batch size B, kernel number E and variance $\sigma$ are determined by allocating 5% of the training data as a validation set and tuning over the intervals $\eta \in [0.0001, 0.01]$, $B \in [64, 512]$, $E \in [1, 7]$ and $\sigma \in [0.01, 10]$. All experiments are conducted on a hardware platform comprising two Intel(R) Xeon(R) Platinum 8383C CPUs (2.70GHz) and an NVIDIA GeForce RTX 4090 GPU.

### B.3 Evaluation metrics

The imputed data matrix $\mathbf{X}^{(\mathrm{imp})}$ is evaluated to assess imputation quality. Following the protocol in [46], we primarily employ the modified mean absolute error (MAE) and root mean squared error (RMSE) for evaluation:

$$\mathrm{MAE} := \frac{1}{\sum_{n=1}^{N} \sum_{d=1}^{D} \bar{m}_{nd}} \sum_{n=1}^{N} \sum_{d=1}^{D} \left| x_{nd}^{(\mathrm{imp})} - x_{nd}^{(\mathrm{obs})} \right| \bar{m}_{nd}, \tag{21}$$

$$\mathrm{RMSE} := \sqrt{ \frac{1}{\sum_{n=1}^{N} \sum_{d=1}^{D} \bar{m}_{nd}} \sum_{n=1}^{N} \sum_{d=1}^{D} \left\| x_{nd}^{(\mathrm{imp})} - x_{nd}^{(\mathrm{obs})} \right\|_{2}^{2} \bar{m}_{nd}}, \tag{22}$$

where $\bar{m}_{nd} \in \bar{\mathbf{M}}$ indicates positions of imputed (originally missing) values with $\bar{m}_{nd} = 1 - m_{nd}$, and $x_{nd}^{(\mathrm{obs})} \in \mathbf{X}^{(\mathrm{obs})}$ is the ground-truth value from the non-missing data. As only the originally missing values are imputed, we restrict the calculation of error metrics to the indices where $\bar{m}_{nd} = 1$.

In addition to the point-wise error metrics above, we also consider the squared Wasserstein distance (abbreviated as WASS) [1], which quantifies the discrepancy between the distributions of the imputed values and the corresponding ground-truth values at the missing positions ($\mathbf{M} = 1$).

## C  Additional experimental results

### C.1  An empirical analysis on complexity

In this section, we examine the practical computational complexity of KPI, typically the computational cost per iteration, which includes both the forward and backward passes. We conduct experiments using Intel® Xeon® Gold 6140 CPUs and Nvidia RTX 4090 GPUs, with each experiment repeated 100 times to ensure reliability.

The results are presented in Fig. 5. **The running time per iteration remains limited (within 4 ms) across a diverse range of hyperparameters**, demonstrating the feasibility of KPI for real-world applications. Other key observations are summarized as follows:

- To explore the impact of batch size (B), we vary B within a wide range from 32 to 1024 while keeping the feature amount (D) to 8. The running cost of the forward pass increased with the batch

Table 4: Imputation performance comparison with missing ratio of 0.1.

| Datasets | BT | | CC | | CBV | | IS | | PK | | QB | | WQW | |
|---|---|---|---|---|---|---|---|---|---|---|---|---|---|---|
| Metrics | MAE | WASS | MAE | WASS | MAE | WASS | MAE | WASS | MAE | WASS | MAE | WASS | MAE | WASS |
| MICE | 0.118 | 0.027 | 0.155 | 0.075 | 0.196 | 0.16 | 0.175 | 0.406 | 0.097 | 0.133 | 0.122 | 0.271 | 0.192 | 0.151 |
| Miss.F | 0.123 | 0.037 | 0.172 | 0.111 | 0.185 | 0.149 | 0.141 | 0.295 | 0.128 | 0.163 | 0.102 | 0.284 | 0.164 | 0.129 |
| Sinkhorn | 0.840 | 0.434 | 0.903 | 0.614 | 0.896 | 0.837 | 0.850 | 2.047 | 0.841 | 1.513 | 0.784 | 2.622 | 0.856 | 0.755 |
| TDM | 0.724 | 0.415 | 0.815 | 0.545 | 0.787 | 0.690 | 0.720 | 1.592 | 0.731 | 1.295 | 0.565 | 1.977 | 0.745 | 0.650 |
| CSDI-T | 0.727 | 1.914 | 0.850 | 2.680 | 0.815 | 3.753 | 0.766 | 16.714 | 0.743 | 12.939 | 0.578 | 20.407 | 0.775 | 4.022 |
| MissDiff | 0.718 | 1.446 | 0.847 | 1.803 | 0.812 | 4.101 | 0.750 | 13.640 | 0.744 | 16.209 | 0.566 | 25.062 | 0.755 | 6.037 |
| GAIN | 0.739 | 0.355 | 0.759 | 0.479 | 0.690 | 0.541 | 0.532 | 1.137 | 0.399 | 0.460 | 0.409 | 1.192 | 0.736 | 0.621 |
| MIRACLE | 0.528 | 0.174 | 0.382 | 0.161 | 0.778 | 0.682 | 3.723 | 26.666 | 3.777 | 18.544 | 0.461 | 1.103 | 0.485 | 0.364 |
| MIWAE | 0.539 | 0.226 | 0.698 | 0.436 | 0.782 | 0.668 | 0.603 | 1.638 | 0.526 | 0.861 | 0.450 | 2.044 | 0.626 | 0.507 |
| Remasker | 0.365 | 0.099 | 1.041 | 0.830 | 0.448 | 0.249 | 0.715 | 1.775 | 0.500 | 0.739 | 0.489 | 1.737 | 0.503 | 0.364 |
| NewImp | 0.383 | 0.091 | 0.273 | 0.110 | 0.231 | 0.101 | 0.423 | 1.013 | 0.251 | 0.281 | 0.305 | 1.067 | 1.045 | 0.834 |
| KPI(Ours) | 0.084 | 0.022 | 0.023 | 0.01 | 0.051 | 0.016 | 0.093 | 0.217 | 0.071 | 0.066 | 0.05 | 0.219 | 0.082 | 0.058 |

Table 5: Imputation performance comparison with missing ratio of 0.2.

| Datasets | BT | | CC | | CBV | | IS | | PK | | QB | | WQW | |
|---|---|---|---|---|---|---|---|---|---|---|---|---|---|---|
| Metrics | MAE | WASS | MAE | WASS | MAE | WASS | MAE | WASS | MAE | WASS | MAE | WASS | MAE | WASS |
| MICE | 0.145 | 0.027 | 0.171 | 0.097 | 0.208 | 0.218 | 0.186 | 0.915 | 0.102 | 0.243 | 0.13 | 0.589 | 0.2 | 0.211 |
| Miss.F | 0.141 | 0.058 | 0.173 | 0.131 | 0.187 | 0.206 | 0.144 | 0.589 | 0.133 | 0.316 | 0.106 | 0.579 | 0.168 | 0.181 |
| Sinkhorn | 0.834 | 0.428 | 0.907 | 0.711 | 0.902 | 1.079 | 0.842 | 3.908 | 0.819 | 2.572 | 0.773 | 5.036 | 0.854 | 1.030 |
| TDM | 0.725 | 0.431 | 0.812 | 0.659 | 0.800 | 0.939 | 0.720 | 3.097 | 0.710 | 2.167 | 0.567 | 3.855 | 0.750 | 0.927 |
| CSDI-T | 0.724 | 1.808 | 0.847 | 2.674 | 0.823 | 3.760 | 0.759 | 15.642 | 0.724 | 12.409 | 0.574 | 19.999 | 0.777 | 4.057 |
| MissDiff | 0.714 | 1.282 | 0.835 | 1.707 | 0.818 | 3.658 | 0.746 | 13.473 | 0.718 | 14.872 | 0.562 | 23.777 | 0.757 | 5.526 |
| GAIN | 0.727 | 0.350 | 0.759 | 0.585 | 0.701 | 0.739 | 0.526 | 2.231 | 0.409 | 0.830 | 0.407 | 2.292 | 0.724 | 0.853 |
| MIRACLE | 0.637 | 0.271 | 0.443 | 0.234 | 0.878 | 1.102 | 3.361 | 43.583 | 3.612 | 31.612 | 0.487 | 2.558 | 0.533 | 0.556 |
| MIWAE | 0.569 | 0.223 | 0.730 | 0.535 | 0.801 | 0.904 | 0.620 | 3.198 | 0.511 | 1.398 | 0.465 | 4.102 | 0.653 | 0.728 |
| Remasker | 0.403 | 0.108 | 0.412 | 1.223 | 0.488 | 0.392 | 0.613 | 2.959 | 0.450 | 1.077 | 0.397 | 2.482 | 0.523 | 0.531 |
| NewImp | 0.441 | 0.141 | 0.360 | 0.201 | 0.310 | 0.221 | 0.411 | 1.937 | 0.283 | 0.592 | 0.329 | 2.273 | 0.468 | 0.265 |
| KPI(Ours) | 0.095 | 0.032 | 0.077 | 0.051 | 0.085 | 0.064 | 0.098 | 0.436 | 0.075 | 0.128 | 0.062 | 0.322 | 0.103 | 0.11 |

size, as expected. This is attributed to the larger matrix inversion in (7), which cannot be efficiently accelerated by GPUs. In contrast, the backward pass cost was weakly correlated with B, since gradient computations after constructing the computation graph can be parallelized.

- To investigate the impact of feature amount ($D$), we maintain a constant batch size of 64. A weak correlation between $D$ and the running time is observed. This is because varying $D$ primarily affects the complexity of each kernel matrix computation, which can be effectively mitigated by GPU acceleration. This highlights a practical advantage of KPI: its efficiency in handling datasets with a large number of features.

- We observe that the type of kernel function also affects the running cost. The gaussian kernel exhibits the largest running time compared to other kernel functions, in terms of both forward pass and the backward pass.

### C.2 Additional overall performance results given different missing ratios

Tab. 4-7 detail the imputation performance of KPI and baselines, with results for different missing ratios: 0.1, 0.2, 0.3, and 0.4 listed separately. The results demonstrate that KPI consistently outperforms the baselines in all settings, achieving superior performance in terms of both MAE and WASS. This consistent superiority across varying missing ratios underscores the effectiveness and robustness of KPI for missing data imputation.

Table 6: Imputation performance comparison with missing ratio of 0.3.

| Datasets | BT | | CC | | CBV | | IS | | PK | | QB | | WQW | |
| Metrics | MAE | WASS | MAE | WASS | MAE | WASS | MAE | WASS | MAE | WASS | MAE | WASS | MAE | WASS |
|---|---|---|---|---|---|---|---|---|---|---|---|---|---|---|
| mice | 0.156 | 0.043 | 0.195 | 0.133 | 0.219 | 0.289 | 0.186 | 1.393 | 0.107 | 0.387 | 0.134 | 1.064 | 0.21 | 0.28 |
| missforest | 0.14 | 0.064 | 0.189 | 0.182 | 0.2 | 0.292 | 0.154 | 1.049 | 0.131 | 0.423 | 0.113 | 1.085 | 0.176 | 0.262 |
| sink | 0.828 | 0.475 | 0.911 | 0.853 | 0.904 | 1.368 | 0.851 | 6.014 | 0.828 | 3.898 | 0.774 | 7.291 | 0.859 | 1.313 |
| tdm | 0.733 | 0.506 | 0.825 | 0.834 | 0.809 | 1.260 | 0.730 | 4.796 | 0.723 | 3.337 | 0.571 | 5.629 | 0.754 | 1.240 |
| CSDI-T | 0.717 | 1.905 | 0.851 | 2.684 | 0.826 | 3.816 | 0.761 | 14.942 | 0.729 | 12.044 | 0.574 | 19.732 | 0.782 | 4.093 |
| MissDiff | 0.718 | 1.317 | 0.842 | 1.656 | 0.822 | 3.313 | 0.751 | 13.341 | 0.725 | 13.806 | 0.563 | 22.714 | 0.759 | 4.894 |
| gain | 0.742 | 0.413 | 0.780 | 0.729 | 0.736 | 1.041 | 0.566 | 3.702 | 0.460 | 1.656 | 0.434 | 3.637 | 0.730 | 1.140 |
| miracle | 0.951 | 0.850 | 0.535 | 0.371 | 0.841 | 1.302 | 3.036 | 54.592 | 3.432 | 43.764 | 0.542 | 4.814 | 0.582 | 0.792 |
| miwae | 0.593 | 0.273 | 0.769 | 0.692 | 0.818 | 1.210 | 0.650 | 4.974 | 0.527 | 2.113 | 0.480 | 5.810 | 0.667 | 0.955 |
| remasker | 0.459 | 0.134 | 0.552 | 0.371 | 0.541 | 0.586 | 0.534 | 3.949 | 0.415 | 1.365 | 0.349 | 2.928 | 0.557 | 0.724 |
| NewImp | 0.481 | 0.181 | 0.472 | 0.341 | 0.423 | 0.443 | 0.442 | 3.066 | 0.321 | 1.015 | 0.350 | 3.666 | 0.558 | 0.379 |
| KPI(Ours) | 0.104 | 0.028 | 0.116 | 0.088 | 0.122 | 0.122 | 0.102 | 0.712 | 0.083 | 0.217 | 0.069 | 0.562 | 0.148 | 0.215 |

Table 7: Imputation performance comparison with missing ratio of 0.4.

| Datasets | BT | | CC | | CBV | | IS | | PK | | QB | | WQW | |
| Metrics | MAE | WASS | MAE | WASS | MAE | WASS | MAE | WASS | MAE | WASS | MAE | WASS | MAE | WASS |
|---|---|---|---|---|---|---|---|---|---|---|---|---|---|---|
| MICE | 0.162 | 0.03 | 0.223 | 0.168 | 0.233 | 0.354 | 0.186 | 1.824 | 0.111 | 0.549 | 0.15 | 1.491 | 0.222 | 0.329 |
| MISS.F | 0.156 | 0.083 | 0.198 | 0.226 | 0.192 | 0.346 | 0.154 | 1.346 | 0.133 | 0.595 | 0.114 | 1.254 | 0.177 | 0.326 |
| Sinkhorn | 0.837 | 0.530 | 0.904 | 1.008 | 0.889 | 1.616 | 0.847 | 7.810 | 0.821 | 4.948 | 0.768 | 9.508 | 0.860 | 1.582 |
| TDM | 0.737 | 0.596 | 0.824 | 1.037 | 0.802 | 1.561 | 0.733 | 6.376 | 0.724 | 4.367 | 0.578 | 7.565 | 0.758 | 1.574 |
| CSDI-T | 0.735 | 1.851 | 0.847 | 2.694 | 0.817 | 3.877 | 0.757 | 14.671 | 0.727 | 11.771 | 0.577 | 19.537 | 0.786 | 4.164 |
| MissDiff | 0.726 | 1.284 | 0.837 | 1.628 | 0.812 | 3.019 | 0.750 | 13.272 | 0.723 | 12.960 | 0.566 | 21.728 | 0.761 | 4.278 |
| GAIN | 0.712 | 0.464 | 0.812 | 0.960 | 0.791 | 1.448 | 0.665 | 6.203 | 0.522 | 2.708 | 0.653 | 11.553 | 0.826 | 1.767 |
| Miracle | 1.066 | 1.401 | 0.602 | 0.483 | 0.826 | 1.529 | 2.714 | 58.421 | 3.250 | 53.215 | 0.595 | 7.425 | 0.620 | 1.027 |
| MIWAE | 0.629 | 0.344 | 0.786 | 0.856 | 0.828 | 1.502 | 0.670 | 6.663 | 0.535 | 2.842 | 0.504 | 7.952 | 0.683 | 1.185 |
| Remasker | 0.528 | 0.182 | 1.022 | 1.541 | 0.636 | 0.860 | 0.534 | 5.653 | 0.424 | 1.891 | 0.368 | 4.096 | 0.601 | 0.926 |
| NewImp | 0.563 | 0.301 | 0.553 | 0.520 | 0.542 | 0.743 | 0.451 | 4.035 | 0.351 | 1.563 | 0.378 | 4.989 | 1.022 | 1.542 |
| KPI(Ours) | 0.113 | 0.039 | 0.132 | 0.134 | 0.144 | 0.191 | 0.107 | 1.022 | 0.09 | 0.337 | 0.084 | 1.028 | 0.159 | 0.303 |

## C.3 Additional overall performance results given different missing mechanisms

Tab. 8 and 9 provide a detailed evaluation of the imputation performance of KPI and various baselines under MAR and MNAR mechanisms, respectively. These missing mechanisms are more complex and challenging compared to the MCAR setting reported in Tab. 1, but they are also more representative of real-world scenarios.

The results demonstrate that KPI consistently outperforms the baselines in all settings, achieving superior performance in terms of both metrics. This consistent superiority across different missing mechanisms highlights the effectiveness and robustness of KPI for missing data imputation, making it a reliable choice for diverse real-world applications.

Table 8: Imputation performance comparison under MAR missing mechanism.

| Datasets | BT | | CC | | CBV | | IS | | PK | | QB | | WQW | |
|---|---|---|---|---|---|---|---|---|---|---|---|---|---|---|
| Metrics | MAE | WASS | MAE | WASS | MAE | WASS | MAE | WASS | MAE | WASS | MAE | WASS | MAE | WASS |
| MICE | 0.481 | 0.109 | 0.626 | 0.349 | 0.839 | 0.652 | 0.677 | 1.113 | 0.492 | 0.946 | 0.604 | 2.42 | 0.824 | 0.775 |
| MISS.F | 0.718 | 0.727 | 0.632 | 0.404 | 0.783 | 0.572 | 0.58 | 1.261 | 0.797 | 1.517 | 0.58 | 2.671 | 0.709 | 0.63 |
| CSDI-T | 1.094 | 5.465 | 0.894 | 3.212 | 0.826 | 4.286 | 0.707 | 15.194 | 1.262 | 19.116 | 0.782 | 23.176 | 0.815 | 4.919 |
| MissDiff | 1.019 | 2.835 | 0.888 | 2.189 | 0.852 | 6.008 | 0.704 | 13.233 | 1.219 | 22.773 | 0.762 | 34.125 | 0.805 | 6.919 |
| gain | 1.082 | 1.187 | 0.782 | 0.570 | 0.700 | 0.503 | 0.456 | 0.609 | 0.709 | 1.383 | 0.552 | 1.716 | 0.729 | 0.672 |
| MIRACLE | 0.699 | 0.510 | 0.356 | 0.141 | 0.710 | 0.530 | 3.837 | 19.874 | 4.518 | 23.842 | 0.605 | 1.636 | 0.494 | 0.385 |
| MIWAE | 0.747 | 0.784 | 0.735 | 0.517 | 0.788 | 0.617 | 0.474 | 0.811 | 0.758 | 1.674 | 0.660 | 2.892 | 0.629 | 0.575 |
| Remasker | 0.598 | 0.689 | 1.023 | 0.854 | 0.467 | 0.235 | 0.691 | 1.149 | 0.668 | 1.062 | 0.557 | 1.563 | 0.489 | 0.397 |
| Sinkhorn | 1.106 | 1.319 | 0.958 | 0.718 | 0.923 | 0.804 | 0.829 | 1.350 | 1.257 | 3.392 | 0.904 | 3.032 | 0.905 | 0.912 |
| TDM | 1.015 | 1.313 | 0.855 | 0.614 | 0.823 | 0.653 | 0.691 | 0.995 | 1.194 | 3.311 | 0.752 | 2.739 | 0.794 | 0.776 |
| NewImp | 0.401 | 0.171 | 0.232 | 0.111 | 0.221 | 0.070 | 0.331 | 0.504 | 0.462 | 0.745 | 0.560 | 3.330 | 0.372 | 0.293 |
| multikip | 0.367 | 0.107 | 0.199 | 0.121 | 0.225 | 0.115 | 0.345 | 0.864 | 0.457 | 0.68 | 0.339 | 1.965 | 0.362 | 0.311 |

Table 9: Imputation performance comparison under MNAR missing mechanism.

| Datasets | BT | | CC | | CBV | | IS | | PK | | QB | | WQW | |
|---|---|---|---|---|---|---|---|---|---|---|---|---|---|---|
| Metrics | MAE | WASS | MAE | WASS | MAE | WASS | MAE | WASS | MAE | WASS | MAE | WASS | MAE | WASS |
| MICE | 0.696 | 0.411 | 0.66 | 0.384 | 0.832 | 0.734 | 0.708 | 1.831 | 0.499 | 0.98 | 0.589 | 2.377 | 0.807 | 0.724 |
| MISS.F | 0.731 | 0.579 | 0.697 | 0.483 | 0.748 | 0.631 | 0.587 | 1.944 | 0.726 | 1.513 | 0.5 | 2.291 | 0.667 | 0.567 |
| CSDI-T | 0.885 | 3.105 | 0.885 | 2.923 | 0.838 | 3.922 | 0.759 | 16.833 | 1.016 | 14.173 | 0.683 | 20.330 | 0.795 | 4.275 |
| GAIN | 0.846 | 0.595 | 0.782 | 0.545 | 0.698 | 0.570 | 0.529 | 1.151 | 0.571 | 1.305 | 0.474 | 1.943 | 0.730 | 0.684 |
| MIRACLE | 0.655 | 0.319 | 0.371 | 0.163 | 0.842 | 0.802 | 3.725 | 27.093 | 4.196 | 25.052 | 0.576 | 2.249 | 0.518 | 0.437 |
| MIWAE | 0.658 | 0.424 | 0.735 | 0.508 | 0.808 | 0.731 | 0.559 | 1.535 | 0.628 | 1.400 | 0.561 | 3.318 | 0.641 | 0.577 |
| Remasker | 0.481 | 0.297 | 1.028 | 0.886 | 0.487 | 0.296 | 0.660 | 1.701 | 0.586 | 1.059 | 0.516 | 2.128 | 0.519 | 0.434 |
| Sinkhorn | 0.967 | 0.752 | 0.940 | 0.698 | 0.925 | 0.911 | 0.854 | 2.121 | 1.049 | 2.906 | 0.844 | 3.755 | 0.880 | 0.859 |
| TDM | 0.858 | 0.732 | 0.849 | 0.620 | 0.821 | 0.756 | 0.724 | 1.640 | 0.969 | 2.713 | 0.661 | 3.202 | 0.772 | 0.744 |
| NewImp | 0.645 | 0.461 | 0.585 | 0.593 | 0.562 | 0.837 | 0.442 | 3.945 | 0.434 | 2.328 | 0.441 | 7.161 | 0.601 | 1.102 |
| multikip | 0.573 | 0.257 | 0.271 | 0.22 | 0.357 | 0.352 | 0.391 | 1.344 | 0.423 | 0.769 | 0.286 | 1.967 | 0.458 | 0.563 |

