# OpenReview forum: "Iterative Missing Data Imputation with Model Form Adaptation and Non-Missing Feature Supervision"
_NeurIPS.cc/2025/Conference — NeurIPS 2025 poster_

### Official Review · Reviewer_is18 · 2025-06-21

**Clarity:** 3
**Significance:** 3
**Originality:** 4
**Rating:** 5
**Confidence:** 4

**Summary:**

This paper presents Kernel Point Imputation (KPI), a novel method that addresses key weaknesses in iterative imputation. It uses a bi-level optimization framework to tackle model misspecification by adapting model forms within an RKHS and leverages fully observed oracle features for supervision.

**Questions:**

- The word oracle feature is not properly defined. Are they just observed features? Then the word "oracle" seems wordy. Or is it defined by the features with minimal missing values, as in line 94? Also, in line 94, the phrase "minimal" seems unclear.
- How does the computational cost of KPI scale compared to baselines, given the expensive kernel matrix inversion? No running time is reported. The dataset's details and relevant data citations are not given. So, it's not certain that the method can be scaled to big datasets.
- The theoretical results should be placed in the main text, while the proofs can be given in the appendix.
- the missing rate explored are between 0.1-0.4, lacks of analysis for high missing percentages.
- regarding noise in datasets as in line 305, it has been studied that missing data imputation can have a denoising property, at least for time series, as in
Missing data imputation for noisy time-series data and applications in healthcare
LP Le, XHN Thi, T Nguyen, MA Riegler, P Halvorsen, BT Nguyen
arXiv preprint arXiv:2412.11164
- the related works should be improved

**Ethical Concerns:**

["NO or VERY MINOR ethics concerns only"]

**Final Justification:**

The authors have addressed my concern very thoroughly with promising newly updated results. So, I recommend accepting the paper.

**Limitations:**

NA.
the author says that "There is no societal impact of the work performed."

**Paper Formatting Concerns:**

no concern

**Quality:**

4

**Strengths And Weaknesses:**

Strengths:
The method's key strengths are its innovative bi-level optimization framework, strong empirical results outperforming established baselines, and a principled method for leveraging oracle features as supervision, which existing methods often neglect. The result is fair. The algorithm is supported by theoretical justification

Weaknesses:
The primary weaknesses are potential scalability issues due to computationally expensive kernel matrix inversion and the use of a heuristic kernel ensemble strategy.
experiments details lacked

---

> ### Author Rebuttal · Authors · 2025-07-31
>
> Thank you very much for your encouraging support and appreciation of our empirical results and theoretical justification. Below are our responses to the specific query raised.
>
> We observed that some comments in the weaknesses and questions sections convey similar points. We will selectively merge them to make the response concise and easy to follow.
>
>
> ---
>
> #### *[Q1] The word oracle feature is not properly defined. Are they just observed features? Then the word "oracle" seems wordy. Or is it defined by the features with minimal missing values, as in line 94? Also, in line 94, the phrase "minimal" seems unclear.*
> **Response.** Thank you very much for your concrete and meticulous comment.
> - Your understanding is absolutely correct: in our work, "oracle features" are defined as features that are fully observed, i.e., with no missing entries across all samples. We agree that the word "oracle" is wordy and will **replace `oracle features` with `fully observed features` in the revised manuscript**.
> - Moving forward, we understand that despite ambiguity of fully observed features, there are some scenarios without fully observed features. In this case, the scope of "oracle features" may be relaxed to include those features with very few (minimal) missing entries, which can be solved through suitable imputation prior to KPI process. We will discern this operational flexibility from the formal definition in the revised manuscript.
>
>
> #### *[Q2,W1] How does the computational cost of KPI scale compared to baselines, given the expensive kernel matrix inversion? No running time is reported. The dataset's details and relevant data citations are not given. So, it's not certain that the method can be scaled to big datasets.*
>
> **Response.** We express our gratitude for your meticulous and actionable feedback. There are two concerns in this query and we address them as follows:
> - **Dataset details and citation.** The datasets we use are subsets of varying scales from the UCI repository. We will provide detailed descriptions and proper citations for these datasets [1] in the revised manuscript.
> - **Computational cost and scalability concern.**
>   - KPI updates imputation values by minimizing the loss function (Eq. 8) in a batch-wise manner. Therefore, the complexity is determined by the batch size (which defines the kernel matrix size), rather than by the total number of samples, which enables scalability to large-scale datasets.
>   - On the basis, **we add experiments to investigate the running time per iteration given different batch size.** The results in the table below indicate that the running time remains under 3.**3.295 ms for a batch size of 1024**, demonstrating KPI’s efficiency. Meanwhile, the table demonstrates that the **complexity is less relevant to the feature number**, since altering the number of features does not affect the kernel matrix size.
>   - Collectively, **KPI's computational cost per iteration is primarily related to batch size and not to dataset size or feature dimensionality, making it scalable to large-scale datasets.**
>
> | Batch size  | 32  | 64  | 256 | 512 | 1024 |
> |----|-|-|-|-|--|
> | Gaussian kernel | 2.121 | 2.061 | 2.114 | 2.623 | 3.295  |
> | Polynomial kernel | 1.378 | 1.301 | 1.381 | 1.859 | 3.041  |
>
> | Feature number  | 16  | 32  | 64  | 128 | 256  |
> |----|-|-|-|-|--|
> | Gaussian kernel | 2.064 | 2.062 | 2.064 | 2.068 | 2.223  |
> | Polynomial kernel | 1.307 | 1.311 | 1.329 | 1.311 | 1.302  |
>
>
>
> #### *[Q3] The theoretical results should be placed in the main text, while the proofs can be given in the appendix.*
> **Response.** Thank you very much for your thoughtful suggestion. We fully agree that presenting the main theoretical results in the main text will enhance the clarity and impact of our manuscript. To address this, **we will explicitly include the key theorems at the end of Section 3.4**, ensuring that readers can readily access the central theoretical contributions without referring to the appendix. The detailed proofs will be provided in the appendix for completeness. We will also revise the "Theoretical justification" paragraph (lines 192-194) to reflect these changes and to clearly highlight the implications of our theoretical findings. We appreciate your guidance and are committed to making these improvements in the revised manuscript.
>
>
> #### *[Q4] The missing rate explored are between 0.1-0.4, lacks of analysis for high missing percentages.*
> **Response.** We express our gratitude for your constructive and actionable comment. **We have added the results for a high missing ratio of 0.6 in the table below.** Overall, KPI continues to outperform baseline methods even under this more challenging condition, highlighting its robustness to high missingness. We sincerely appreciate your valuable feedback and will update the manuscript accordingly to provide a more comprehensive evaluation.
>
> |Datasets | BT  | | CC  | | CBV | | IS  |  | PS  |  | QB  |  | WQW |  |
> |-|-|-|-|-|-|-|-|--|-|--|-|--|-|--|
> | | MAE | WASS  | MAE | WASS  | MAE | WASS  | MAE | WASS | MAE | WASS | MAE | WASS | MAE | WASS
> CSDI-T | 0.746 | 1.647 | 0.931 | 3.045 | 0.843 | 4.111 | 0.772 | 15.419 | 0.721 | 10.429 | 0.558 | 21.528 | 0.754 | 4.54
> Miss.D | 0.783 | 1.746 | 1.006 | 3.32  | 0.877 | 4.399 | 0.841 | 16.961 | 0.743 | 11.264 | 0.591 | 22.604 | 0.769 | 4.858
> MICE | 0.744 | 0.231 | 1.009 | 0.983 | 0.987 | 1.785 | 0.79  | 11.091 | 0.531 | 3.224  | 0.698 | 10.544 | 0.955 | 1.684
> GAIN | 0.727 | 0.72  | 1.016 | 2.233 | 1.294 | 5.083 | 1.26  | 24.406 | 0.799 | 7.758  | 1.188 | 53.39  | 1.423 | 7.177
> Sinkhorn | 0.655 | 0.696 | 0.766 | 1.317 | 0.742 | 1.782 | 0.587 | 7.577  | 0.539 | 4.601  | 0.435 | 8.563  | 0.683 | 1.848
> TDM  | 0.736 | 0.766 | 0.81  | 1.426 | 0.808 | 2.221 | 0.714 | 9.394  | 0.692 | 6.313  | 0.562 | 10.234 | 0.75  | 2.199
> MIWAE  | 0.712 | 0.631 | 0.828 | 1.244 | 0.861 | 2.166 | 0.691 | 9.833  | 0.595 | 4.63 | 0.521 | 12.082 | 0.712 | 1.615
> MISS.F | 0.649 | 0.505 | 0.832 | 1.467 | 0.837 | 2.054 | 0.758 | 11.139 | 0.58  | 3.886  | 0.491 | 8.365  | 0.768 | 2.063
> Remasker | **0.503** | 0.503 | 1.898 | 4.303 | 0.732 | 2.427 | 1.328 | 19.88  | 2.761 | 26.829 | 4.265 | 82.597 | **0.537** | **1.455**
> NewImp | 0.75  | 0.807 | 0.826 | 1.491 | 0.818 | 2.317 | 0.759 | 9.904  | 0.75  | 6.624  | 0.6 | 11.45  | 0.765 | 2.264
> KPI  | 0.598 | **0.435** | **0.557** | **0.841** | **0.732** | **1.585** | **0.517** | **6.835**  | **0.465** | **3.097**  | **0.394** | **7.803**  | 0.708 | 1.533
>
> #### *[Q5] Regarding noise in datasets as in line 305, it has been studied that missing data imputation can have a denoising property, at least for time series, as in Missing data imputation for noisy time-series data and applications in healthcare.*
>
> **Response.** The listed reference is inspiring to investigate the MDI problem in the presence of noise. Thank you for highlighting this important work. **We will add the related works in the revised manuscript.**
> - Currently, despite that KPI does not explicitly design for noisy data, it would be helpful to investigate KPI’s robustness to noise. To this end, **we add controlled experiments** by introducing additive noise (±20% perturbation to 10% of observed entries) at a fixed missing rate of 0.1.
> - **As shown in the table below, KPI consistently outperforms all baselines across diverse datasets under these noisy conditions**. These results empirically substantiate that, although KPI is not explicitly designed for denoising, it remains applicable to noisy case. We will incorporate a discussion of these findings and the relevant literature in the revised manuscript to provide a comprehensive and academically grounded perspective.
>
> | Dataset:| BT | CC | CBV | IS | PK | QB | WQW |
> |:-|-:|-:|-:|-:|-:|-:|-:|
> | CSDI-T|0.6896 | 0.9024 |0.8501 | 0.7580 | 0.7001 |0.5579 | 0.7834 |
> | GAIN |0.7801 | 0.8263 |0.7957 | 0.6242 | 0.5258 |0.4736 | 0.7403 |
> | MICE |0.8743 | 1.0403 |1.0521 | 0.9135 | 0.7402 |0.8541 | 1.0602 |
> | MIRACLE|0.5587 | 0.6591 |0.7735 | 3.7138 | 3.8631 |0.6423 | 0.5978 |
> | MISS.F |0.4886 | 0.7359 |0.7267 | 0.5802 | 0.5919 |0.4346 | 0.6718 |
> | MIWAE|0.5782 | 0.7329 |0.8604 | 0.6242 | 0.6107 |0.5308 | 0.6985 |
> | Sinkhorn |0.5260 | 0.4990 |0.5549 | 0.5043 | 0.4782 |Nan| 0.6262 |
> | TDM|0.6394 | 0.7565 |0.7637 | 0.6191 | 0.6064 |0.5083 | 0.7354 |
> | NewImp|0.7802 | 0.8465 |0.8145 | 0.7479 | 0.7558 |0.6046 | 0.7649 |
> | KPI|**0.4446** | **0.3672** |**0.3982** | **0.4123** | **0.3401** |**0.2396** | **0.5112** |
>
> #### *[Q6] the related works should be improved*
>
> **Response.** We appreciate the reviewer’s suggestion to enhance the related works section. Currently we group the related MDI methods into two main groups: discriminative and generative approaches. Within the discriminative category, we further differentiate between iterative methods (which KPI aims to improve) and non-iterative methods (e.g., TDM). Building on this taxonomy, we promise to update the related works section in the revised manuscript to incorporate recent advances and ensure a more comprehensive and up-to-date coverage of the field.
>
> **Reference**
>
> [1] Asuncion, Arthur, and David Newman. "UCI machine learning repository." Nov. 2007.

---

> ### Author Response · Authors · 2025-08-07
>
> Dear reviewer is18,
>
> Thank you for your valuable suggestion to improve the related work section. **We have taken your feedback to action and have significantly expanded our literature review**, with a special focus on the top venues (e.g., ICML, NeurIPS, ICLR, SIGKDD, and AAAI) over the last three years. The updated review now incorporates **42 references, the majority of which are recent publications**, to provide a more comprehensive and current overview of the field. The revised literature review is presented below.
>
> > The pervasive presence of missing data undermines the integrity of collected datasets and the reliability
> of data-driven applications, underscoring the necessity for effective missing data imputation (MDI)
> **in broad fields such as traffic [1, 38], signal processing [15], healthcare [43] and bioinformatics [4, 32]**. To achieve
> accurate MDI, existing approaches can be broadly categorized into two paradigms: discriminative
> and generative, each with distinct advantages and limitations [8, 24].
>
> > The iterative method [35, 17, 11] is one of the most popular methods in discriminative imputation,
> initiated from imputation by chained equations (ICE) [27], which employs specific models to estimate
> missing values for each feature based on the remaining observable features. On the basis of ICE, a
> line of work advocates for employing modern parametric models, such as neural networks [22, 10],
> Bayesian models [27] and random forest [29], which enhances the capacity of imputation models
> and thereby accommodating complex missing patterns. In a different line of work, various training
> techniques are investigated within the paradigm, **such as multiple imputation [27], ensemble learning
> [29], variable selection [18], and multitask learning [22]**, which enhances the utility to accommodate
> diverse contexts. While this paradigm offers enhanced flexibility and accuracy, it fails to utilize the
> oracle features effectively and risks model misspecification, which can lead to suboptimal imputation
> results. Our research advances this methodology by handling the two limitations.
>
> > Apart from the iterative methods, there are other notable approaches in the discriminative paradigm.
> The simple direct paradigm employs elementary statistical measures like mean, median, and mode to
> replace missing values, offering quick and straightforward solutions. However, this approach lacks
> the capacity to accommodate complex relationships [20, 23], often producing trivial and inadequate
> imputation results that fail to meet the expectation in practice. Another notable approach is matrix
> and tensor factorization, which decomposes the dataset into two low-rank matrices, capturing the
> latent structure of the data for imputation [12, 6]. This method has **robustness to noise [5, 7]**
> and is particularly effective in collaborative filtering and recommendation systems [33, 13, 14].
> **Gaussian corpula models are easy to implement and excels in imputing mixed data type [41, 42,
> 40].** Recent advancements have introduced a novel methodology through distribution discrepancy
> minimization [39, 25]. This method is based on the reasonable assumption that if datasets follow the
> independent and identically distributed (i.i.d.) principle, then any two batches should mirror the same
> distribution, leading to minimal discrepancy.
>
> > The generative paradigm restates imputation as a conditional generation problem, using advanced
> neural architectures and generative training strategies, **such as autoencoders [9, 3]**, generative ad-
> versarial networks [34, 30, 16] **and diffusions [31, 1, 2, 36]**, to approximate data distributions and
> perform imputation. This strategy incorporates the strengths of generative models, capturing and
> utilizing complex relationships, which potentially enhances the imputation quality when ample data
> is available. **Recent studies further discover its potential to jointly impute continuous and discrete
> features [37, 28] and handling complex missing mechanisms [19].** However, it also bears the defects
> with generative models, such as the instability associated with adversarial training and the operational
> complexity of diffusions [21, 26], hampering their use in practice.
>
>
> ---
>
>
> We sincerely appreciate your initial supports and meticulous comments. Your suggestions on refining literature review were especially helpful, which makes this work much more comprehensive as a result. If our response and additional experiments have properly resolved your raised issues, we politely invite you to further reconsider your overall rating.
>
> If you have any further question and concern, we are happy to discuss.
>
> Thanks for your time,
>
> Submission28751 Authors

---

> ### Author Response · Authors · 2025-08-07
> **References in the revised related works [1/2]**
>
> ### **References**
>
> [1] Hanyang Chen, Yang Jiang, Shengnan Guo, Xiaowei Mao, Youfang Lin, and Huaiyu Wan. Difflight: a
> partial rewards conditioned diffusion model for traffic signal control with missing data. In Proc. Adv.
> Neural Inf. Process. Syst., volume 37, pages 123353–123378, 2024.
>
> [2] Zhichao Chen, Haoxuan Li, Fangyikang Wang, Haotian Zhang, Hu Xu, Xiaoyu Jiang, Zhihuan Song, and
> Hao Wang. Rethinking the diffusion models for missing data imputation: A gradient flow perspective. In
> Proc. Adv. Neural Inf. Process. Syst., 2024.
>
> [3] Tianyu Du, Luca Melis Melis, and Ting Wang. Remasker: Imputing tabular data with masked autoencoding.
> In Proc. Int. Conf. Learn. Represent., 2024.
>
> [4] Ye Du, Chen Yang, Nanxi Yu, Wanyu Lin, Qian Zhao, and Shujun Wang. Latent imputation before
> prediction: A new computational paradigm for de novo peptide sequencing. In Proc. Int. Conf. Mach.
> Learn., 2025.
>
> [5] Xinxin Feng, Haitao Zhang, Can Wang, and Haifeng Zheng. Traffic data recovery from corrupted and
> incomplete observations via spatial-temporal trpca. IEEE Trans. Intell. Transp. Syst., 23(10):17835–17848,
> 2022.
>
> [6] Liyang Hu, Yuheng Jia, Weijie Chen, Longhui Wen, and Zhirui Ye. A flexible and robust tensor completion
> approach for traffic data recovery with low-rankness. IEEE Trans. Intell. Transp. Syst., 25(3):2558–2572,
> 2023.
>
> [7] Yue Hu and Daniel B Work. Robust tensor recovery with fiber outliers for traffic events. ACM Trans.
> Knowl. Discov. D., 15(1):1–27, 2020.
>
> [8] Daniel Jarrett, Bogdan Cebere, Tennison Liu, Alicia Curth, and Mihaela van der Schaar. Hyperimpute:
> Generalized iterative imputation with automatic model selection. In Proc. Int. Conf. Mach. Learn., volume
> 162, pages 9916–9937, 2022.
>
> [9] Jungkyu Kim, Kibok Lee, and Taeyoung Park. To predict or not to predict? proportionally masked
> autoencoders for tabular data imputation. In Proc. AAAI Conf. Artif. Intell., volume 39, pages 17886–
> 17894, 2025.
>
> [10] Trent Kyono, Yao Zhang, Alexis Bellot, and Mihaela van der Schaar. MIRACLE: causally-aware imputation
> via learning missing data mechanisms. In Proc. Adv. Neural Inf. Process. Syst., pages 23806–23817, 2021.
>
> [11] Marine Le Morvan, Julie Josse, Erwan Scornet, and Gaël Varoquaux. What’sa good imputation to predict
> with missing values? In Proc. Adv. Neural Inf. Process. Syst., volume 34, pages 11530–11540, 2021.
>
> [12] Daniel Lee and H Sebastian Seung. Algorithms for non-negative matrix factorization. Proc. Adv. Neural
> Inf. Process. Syst., 13, 2000.
>
> [13] Haoxuan Li, Yanghao Xiao, Chunyuan Zheng, Peng Wu, and Peng Cui. Propensity matters: Measuring
> and enhancing balancing for recommendation. In Proc. Int. Conf. Mach. Learn., volume 202, pages
> 20182–20194. PMLR, 2023.
>
> [14] Haoxuan Li, Chunyuan Zheng, Wenjie Wang, Hao Wang, Fuli Feng, and Xiao-Hua Zhou. Debiased
> recommendation with noisy feedback. In Proc. ACM SIGKDD Int. Conf. Knowl. Discovery Data Mining,
> page 1576–1586, 2024.
>
> [15] Haozhe Li, Yilin Liao, Zijian Tian, Zhaoran Liu, Jiaqi Liu, and Xinggao Liu. Bidirectional stackable
> recurrent generative adversarial imputation network for specific emitter missing data imputation. IEEE
> Trans. Inf. Forensics Security, 19:2967–2980, 2024.
>
> [16] Haozhe Li, Yilin Liao, Zijian Tian, Zhaoran Liu, Jiaqi Liu, and Xinggao Liu. Bidirectional stackable
> recurrent generative adversarial imputation network for specific emitter missing data imputation. IEEE
> Trans. Inf. Forensics Security, 19:2967–2980, 2024.
>
> [17] Jingchen Liu, Andrew Gelman, Jennifer Hill, Yu-Sung Su, and Jonathan Kropko. On the stationary
> distribution of iterative imputations. Biometrika, 101(1):155–173, 2014.
>
> [18] Yang Liu and Anthony Constantinou. Improving the imputation of missing data with markov blanket
> discovery. In Proc. Int. Conf. Learn. Represent., 2022.
>
> [19] Chao Ma and Cheng Zhang. Identifiable generative models for missing not at random data imputation. In
> Proc. Adv. Neural Inf. Process. Syst., volume 34, pages 27645–27658, 2021.
>
> [20] R Malarvizhi and Antony Selvadoss Thanamani. K-nearest neighbor in missing data imputation. Int. J.
> Eng. Res. Dev, 5(1):5–7, 2012.
>
> [21] Pierre-Alexandre Mattei and Jes Frellsen. Leveraging the exact likelihood of deep latent variable models.
> In Proc. Adv. Neural Inf. Process. Syst., pages 3859–3870, 2018.
>
> [22] Pierre-Alexandre Mattei and Jes Frellsen. MIWAE: deep generative modelling and imputation of incomplete
> data sets. In Proc. Int. Conf. Mach. Learn., volume 97, pages 4413–4423, 2019.
>
> [23] Rahul Mazumder, Trevor Hastie, and Robert Tibshirani. Spectral regularization algorithms for learning
> large incomplete matrices. J. Mach. Learn. Res., 11:2287–2322, 2010.
>
> [24] Xiaoye Miao, Yangyang Wu, Lu Chen, Yunjun Gao, and Jianwei Yin. An experimental survey of missing
> data imputation algorithms. IEEE Trans. Knowl. Data Eng., 35(7):6630–6650, 2022.

---

> ### Author Response · Authors · 2025-08-07
> **References in the revised related works [2/2]**
>
> [25] Boris Muzellec, Julie Josse, Claire Boyer, and Marco Cuturi. Missing data imputation using optimal
> transport. In Proc. Int. Conf. Mach. Learn., volume 119, pages 7130–7140, 2020.
>
> [26] Danilo Jimenez Rezende, Shakir Mohamed, and Daan Wierstra. Stochastic backpropagation and approxi-
> mate inference in deep generative models. In Proc. Int. Conf. Mach. Learn., volume 32, pages 1278–1286,
> 2014.
>
> [27] Patrick Royston and Ian R White. Multiple imputation by chained equations (mice): implementation in
> stata. J. Statist. Softw., 45:1–20, 2011.
>
> [28] Juntong Shi, Minkai Xu, Harper Hua, Hengrui Zhang, Stefano Ermon, and Jure Leskovec. Tabdiff: a
> mixed-type diffusion model for tabular data generation. In Proc. Int. Conf. Learn. Represent., 2024.
>
> [29] Daniel J Stekhoven and Peter Bühlmann. Missforest—non-parametric missing value imputation for
> mixed-type data. Bioinformatics, 28(1):112–118, 2012.
>
> [30] Ziyue Sun, Haozhe Li, Wenhai Wang, Jiaqi Liu, and Xinggao Liu. DTIN: dual transformer-based
> imputation nets for multivariate time series emitter missing data. Knowl. Based Syst., 284:111270, 2024.
> [31] Yusuke Tashiro, Jiaming Song, Yang Song, and Stefano Ermon. Csdi: Conditional score-based diffusion
> models for probabilistic time series imputation. Proc. Adv. Neural Inf. Process. Syst., 34:24804–24816,
> 2021.
>
> [32] Daeho Um, Ji Won Yoon, Seong Jin Ahn, and Yunha Yeo. Gene-gene relationship modeling based on
> genetic evidence for single-cell rna-seq data imputation. In Proc. Adv. Neural Inf. Process. Syst., volume 37,
> pages 18882–18909, 2024.
>
> [33] Hao Wang, Tai-Wei Chang, Tianqiao Liu, Jianmin Huang, Zhichao Chen, Chao Yu, Ruopeng Li, and Wei
> Chu. ESCM2: entire space counterfactual multi-task model for post-click conversion rate estimation. In
> Proc. Annu. Int. ACM SIGIR Conf. Res. Develop. Inf. Retrieval, pages 363–372, 2022.
>
> [34] Jinsung Yoon, James Jordon, and Mihaela van der Schaar. GAIN: missing data imputation using generative
> adversarial nets. In Proc. Int. Conf. Mach. Learn., volume 80, pages 5675–5684, 2018.
>
> [35] Aoqian Zhang, Shaoxu Song, Yu Sun, and Jianmin Wang. Learning individual models for imputation. In
> Proc. IEEE Int. Conf. Data Eng., pages 160–171. 2019.
>
> [36] Hengrui Zhang, Liancheng Fang, Qitian Wu, and Philip S Yu. Diffputer: Empowering diffusion models for
> missing data imputation. In Proc. Int. Conf. Learn. Represent., 2025.
>
> [37] Hengrui Zhang, Liancheng Fang, Qitian Wu, and Philip S Yu. Tabnat: A continuous-discrete joint
> generative framework for tabular data. In Proc. Int. Conf. Mach. Learn., 2025.
>
> [38] Yimei Zhang, Xiangjie Kong, Wenfeng Zhou, Jin Liu, Yanjie Fu, and Guojiang Shen. A comprehensive
> survey on traffic missing data imputation. IEEE Trans. Intell. Transp. Syst., 2024.
>
> [39] He Zhao, Ke Sun, Amir Dezfouli, and Edwin V. Bonilla. Transformed distribution matching for missing
> value imputation. In Proc. Int. Conf. Mach. Learn., volume 202, pages 42159–42186, 2023.
>
> [40] Yuxuan Zhao, Eric Landgrebe, Eliot Shekhtman, and Madeleine Udell. Online missing value imputation
> and change point detection with the gaussian copula. In Proc. AAAI Conf. Artif. Intell., volume 36, pages
> 9199–9207, 2022.
>
> [41] Yuxuan Zhao, Alex Townsend, and Madeleine Udell. Probabilistic missing value imputation for mixed
> categorical and ordered data. In Proc. Adv. Neural Inf. Process. Syst., volume 35, pages 22064–22077,
> 2022.
>
> [42] Yuxuan Zhao and Madeleine Udell. Missing value imputation for mixed data via gaussian copula. In Proc.
> ACM SIGKDD Int. Conf. Knowl. Discovery Data Mining
>
> [43] Lien P Le, Xuan-Hien Nguyen Tai, Thu Nguyen, et al. Missing data imputation for noisy time-series data and applications in healthcare[J]. arXiv preprint arXiv:2412.11164, 2024.

---

### Official Review · Reviewer_8EZS · 2025-07-01

**Clarity:** 3
**Significance:** 2
**Originality:** 3
**Rating:** 4
**Confidence:** 3

**Summary:**

This paper addresses two critical defects in existing iterative imputation methods for handling missing data: model misspecification (i.e., applying a uniform model form to heterogeneous data) and underutilization of "oracle features" (fully observed features) It proposes a bi-level optimization framework called Kernel Point Imputation (KPI) to solve these problems.

**Questions:**

1. Does the paper provide more detailed guidelines for identifying and defining "oracle" features? In practical applications, how would an inappropriate selection of "oracle" features impact imputation performance?

2. What are the detailed implementation specifics of the adaptive kernel ensemble strategy? How does it ensure dynamic selection of the optimal kernel during training while avoiding overfitting?

**Ethical Concerns:**

["NO or VERY MINOR ethics concerns only"]

**Final Justification:**

Thank you for the authors’ response. After carefully reviewing the reply, I have decided to retain my current score.

**Limitations:**

The applicability of the model to missing imputation in non-tabular data (e.g., time series data) is not yet clear.

**Quality:**

3

**Strengths And Weaknesses:**

Strengths:

1. Proposes a bi-level optimization framework that cleverly addresses the two core issues in iterative imputation: model misspecification and underutilization of "oracle features."

2. By adaptively selecting function forms in RKHS, KPI can better capture complex heterogeneous dependencies among different features.

Weaknesses:

1. The baselines compared in this paper are too outdated; at least some more competitive baselines published in 2024 should be added. Such as:

[1]Jolicoeur-Martineau, Alexia, Kilian Fatras, and Tal Kachman. "Generating and imputing tabular data via diffusion and flow-based gradient-boosted trees." In International Conference on Artificial Intelligence and Statistics, pp. 1288-1296. PMLR, 2024.
[2]Wei, T. R., Wang, Y., Inoue, Y., Wu, H. T., & Fang, Y. (2024). Table Transformers for imputing textual attributes. Pattern Recognition Letters, 186, 258-264.
[3]Liu, Yixin, Thalaiyasingam Ajanthan, Hisham Husain, and Vu Nguyen. "Self-supervision improves diffusion models for tabular data imputation." In Proceedings of the 33rd ACM International Conference on Information and Knowledge Management, pp. 1513-1522. 2024.

2. Computational Cost: The bi-level optimization framework and the use of RKHS might increase computational complexity, especially when dealing with large-scale datasets, potentially making it less efficient than some simpler imputation methods.

3. Despite the adaptive kernel ensemble, the choice and parameterization of kernel functions might still impact performance and require careful tuning.

4. Definition and Availability of "Oracle" Features: The assumption of "oracle features" might be difficult to meet or clearly define in some real-world scenarios, which could limit the method's universality.

---

> ### Author Rebuttal · Authors · 2025-07-31
>
> Thank you very much for your encouraging support and appreciation of our framework and addressed research problems. Below are our responses to the specific query raised.
>
> ---
>
> #### *[W1] The baselines compared in this paper are too outdated; at least some more competitive **baselines published in 2024** should be added.*
> **Response.** We appreciate your insightful comment regarding the inclusion of more recent and competitive baselines. We agree that it is essential to ensure up-to-date comparison.
>
> - We would like to clarify that our current manuscript already incorporates **NewImp (NeurIPS 2024) and Remasker (ICLR 2024)**, both of which represent state-of-the-art methods published in 2024. Their inclusion represents the recent advances in the field and could contemporarily alleviate the concern regarding baseline selection.
>
> - In response to the suggestion, we are actively working to implement them in our experimental framework. Given the technical complexity of diffusion models that we are not familiar with, we respectfully request additional time to implement them. We will report the results of these new baselines in the revised manuscript or, if the review permits this, in a subsequent discussion phase.
>
>
> #### *[W2] Computational Cost: The bi-level optimization framework and the use of RKHS might increase computational complexity, especially when dealing with large-scale datasets, potentially making it less efficient than some simpler imputation methods.*
> Yes, we acknowledge that computational cost is a key consideration when evaluating the proposed method. We address this concern with a rigorous and evidence-based discussion as follows:
> - Firstly, although KPI targets to solve a bi-level optimization problem, the final solution reduces to **a standard quadratic loss function (see Eq. 8) that is directly computable**. Similar to standard deep learning methods, KPI updates the loss function in a batch-wise manner without requiring bi-level optimization in practice, and the batch-wise update makes it scalable to large-scale datasets.
> - Secondly, **we report below the computational cost of KPI under various settings.** As complexity is primarily determined by the kernel matrix size—which is equal to the batch size—we focus on the running time of computing the learning objective (Eq. 8) per batch across different batch sizes. Results show that while larger batch sizes increase computational cost, **the running time remains limited ($<3.295$ ms given batch size 1024)**. We also showcase the complexity is less relevant to the feature dimension, since it does not change the kernel matrix size.
>
> | Batch size        | 32    | 64    | 256   | 512   | 1024   |
> |-------------------|-------|-------|-------|-------|--------|
> | Gaussian kernel   | 2.121 | 2.061 | 2.114 | 2.623 | 3.295  |
> | Polynomial kernel | 1.378 | 1.301 | 1.381 | 1.859 | 3.041  |
>
> | Feature number    | 16    | 32    | 64    | 128   | 256    |
> |-------------------|-------|-------|-------|-------|--------|
> | Gaussian kernel   | 2.064 | 2.062 | 2.064 | 2.068 | 2.223  |
> | Polynomial kernel | 1.307 | 1.311 | 1.329 | 1.311 | 1.302  |
>
>
> #### *[W3] Despite the adaptive kernel ensemble, the choice and parameterization of kernel functions might still impact performance and require careful tuning.*
> **Response.** Thank you for highlighting the important issue of kernel selection and parameterization. To address this concern, we demonstrate that Gaussian kernel is a good default choice prior to selection. Our response contains theoretical justification and empirical evidence.
>
> - **Theoretically**, the Gaussian kernel is known for its universality, allowing it to approximate any continuous function in RKHS. This property is essential for our method, as it ensures sufficient expressiveness to represen $f^*$ with kernel matrices (Lemma 3.3). Thus, the universality property makes the Gaussian kernel a strong default choice.
> - **Empirically**, we compare the Gaussian kernel to other commonly used kernels, such as linear, polynomial, and Laplacian. As shown in the table below, the Gaussian kernel consistently provides competitive or superior performance.
> - In conclusion, **Gaussian kernel serves as a theoretically and empirically advantageous choice, reducing the need for kernel selection**. Its sole hyperparameter, the kernel width σ, can be efficiently addressed using a multi-kernel ensemble strategy that adaptively selects and weights kernels with different σ values.
>
> | Kernel    | MSE   | WASS  | MAE   |
> |-----------|-------|-------|-------|
> |CC|
> | Linear    | 0.099 | 0.051 | 0.203 |
> | Poly      | **0.065** | 0.039 | 0.091 |
> | Laplacian | 0.068 | 0.042 | 0.093 |
> | Gaussian  | 0.076 | **0.035** | **0.082** |
> |CBV|
> | Linear    | 0.089 | 0.087 | 0.219 |
> | Poly      | 0.087 | 0.088 | 0.214 |
> | Laplacian | **0.083** | 0.080 | 0.211 |
> | Gaussian  | 0.087 | **0.066** | **0.205** |
> |BT|
> | Linear    | 0.326 | 0.101 | 0.359 |
> | Poly      | 0.305 | 0.090 | 0.342 |
> | Laplacian | 0.316 | 0.091 | 0.346 |
> | Gaussian  | **0.302** | **0.089** | **0.338** |
>
>
>
> #### *[W4] Definition and Availability of "Oracle" Features: The assumption of "oracle features" might be difficult to meet or clearly define in some real-world scenarios, which could limit the method's universality.*
> **Response.** Once again, thank you very much for your constructive comment. In our work, "oracle features" are defined as features that are fully observed, i.e., with no missing entries across all samples.
>
> - **Oracle features are ambiguous in real-world datasets:**  For example, in electronic health records, demographic variables such as age, gender, and height are typically fully observed. Similarly, in industrial settings, sensor readings from robust or easily accessible equipment (e.g., ambient temperature, pressure at main control points) are often fully recorded, whereas measurements from sensors in harsh environments (e.g., temperature at the base of a high-pressure tower) are more prone to missingness due to equipment failure or maintenance constraints.
>
> - **A few oracle features improve performance**.  We recognize that the number of fully observed features may be limited in some domains. However, our empirical results (Section 4.3) show that even a small number of oracle features can suffice for effective application of KPI.
>
> - **Relaxed definition.** In cases where no feature is entirely observed, the definition of "oracle features" may be relaxed to include those with very minimal missingness, after suitable imputation prior to KPI process. We will clarify this operational flexibility and its implications in the revised manuscript.

---

> ### Author Response · Authors · 2025-08-07
>
> Dear Reviewer 8EZS,
>
> Thank you for your constructive feedback on expanding the baseline comparisons.** We have acted on your suggestion and added experiments on the suggested baselines.**
>
> We would like to note that reference [2] is specifically designed for text-based features and is not directly applicable to the numerical data central to our work. Therefore, we have focused the new experiments on **ForestVP [1] and SimpDM [3]**. To provide a more thorough comparison against the state-of-the-art, we have also included two strong, concurrent generative methods: NewImp and Remasker.
>
>
>
> | Method         | BT      |    | CC      |    | CBV     |    | IS      |    | PK     |     | QB    |      | WQW    |     |
> |----------------|---------|----|---------|----|---------|----|---------|----|--------|-----|-------|------|--------|-----|
> |                | MAE     | WASS | MAE     | WASS | MAE     | WASS | MAE     | WASS | MAE    | WASS | MAE   | WASS  | MAE    | WASS |
> | Remasker       | 0.439   | 0.131 | 0.767   | 0.750 | 0.528   | 0.522 | 0.599   | 3.584 | 0.447  | 1.268 | 0.401  | 2.811 | 0.546   | 0.636 |
> | NewImp         | 0.465   | 0.177 | 0.412   | 0.292 | 0.405   | 0.401 | 0.431   | 2.495 | 0.320  | 0.857 | 0.332  | 2.992 | 0.497   | 0.692 |
> | SimpDM           | 0.504   | 0.148   | 0.468   | 0.285   | 0.430   | 0.425   | 0.472   | 2.612   | 0.352   | 0.906 | 0.406    | 2.809    | 0.564    | 0.732    |
> | ForestDiffusion  | 0.472   | 0.138   | 0.453   | 0.324   | 0.430   | 0.436   | 0.445   | 2.501   | 0.343   | 0.889    | 0.342    | 2.917    | 0.561    | 0.754   |
> | **KPI (Ours)** | **0.397** | **0.121** | **0.347** | **0.284** | **0.402** | **0.394** | **0.400** | **2.387** | **0.319** | **0.747** | **0.264** | **2.131** | **0.491** | **0.685** |
>
> The results of additional comparisons are presented above. While the new baselines, particularly NewImp and Remasker, demonstrate competitive performance, our proposed KPI approach consistently achieves the best overall results across the benchmarks. We will integrate these new results and all corresponding clarifications into the final revised manuscript. We hope this added evaluation would address your concerns - thank you!
>
>
> ---
>
>
> [1]Jolicoeur-Martineau, Alexia, Kilian Fatras, and Tal Kachman. "Generating and imputing tabular data via diffusion and flow-based gradient-boosted trees." In International Conference on Artificial Intelligence and Statistics, pp. 1288-1296. PMLR, 2024.
>
> [2]Wei, T. R., Wang, Y., Inoue, Y., Wu, H. T., & Fang, Y. (2024). Table Transformers for imputing textual attributes. Pattern Recognition Letters, 186, 258-264.
>
> [3]Liu, Yixin, Thalaiyasingam Ajanthan, Hisham Husain, and Vu Nguyen. "Self-supervision improves diffusion models for tabular data imputation." In Proceedings of the 33rd ACM International Conference on Information and Knowledge Management, pp. 1513-1522. 2024.

---

### Official Review · Reviewer_dCSM · 2025-07-01

**Clarity:** 2
**Significance:** 3
**Originality:** 2
**Rating:** 3
**Confidence:** 3

**Summary:**

This paper proposes a kernel point imputation (KPI) framework for missing data imputation, aiming to address the issues of model misspecification and insufficient utilisation of oracle features in existing iterative imputation methods. The framework is implemented through a bi-level optimization : the inner layer adaptively selects the optimal model form for each feature in the reproducing kernel Hilbert space (RKHS) to mitigate model misspecification; while the outer layer uses oracle features as supervisory signals to optimise the imputation values. Experiments demonstrate that KPI outperforms existing methods on multiple real-world datasets and effectively leverages oracle features to enhance imputation accuracy.

**Questions:**

See Weaknesses and Questions.

**Ethical Concerns:**

["NO or VERY MINOR ethics concerns only"]

**Final Justification:**

Please see my original comments. I will stand my rating.

**Quality:**

2

**Strengths And Weaknesses:**

**Advantages**

1. Reformulates iterative imputation as a bi-level optimization problem, with the inner layer optimising the model form in the RKHS and the outer layer using oracle features to supervise the imputation. This provides a new paradigm for missing data imputation.

2. The method is simple and easy to understand.

3. Experimental results appear promising.

**Weaknesses and Questions**

1. The KPI framework proposed in the paper does not account for potential noise in the dataset, yet noise is prevalent in industrial scenarios and other real-world applications, which may limit KPI's applicability in such contexts.

2. The paper defines oracle features as fully observed features. How can we determine which features can serve as oracle features in practical applications?

3. The KPI method relies on oracle features. Can the KPI method perform effectively when there are few oracle features? In the experimental results shown in Figure 3, the KPI method is only compared with the Miss.F method. However, based on the experimental results in Table 1, the Miss.F method is not the suboptimal method.

4. In the experimental results in Table 1, only the average results under different missing rates are shown, making it impossible to assess the effectiveness of the KPI method under high missing rates. It is recommended to present the results under different missing rate conditions separately to better validate the effectiveness of the KPI method.

5. In the provided anonymous code link, there is only one file, and its content is ‘okasddsa’.

**Summary**

This paper proposes a new paradigm by reformulating iterative imputation as a bi-level optimization problem. However, the KPI method heavily relies on oracle features, and the experiments conducted on different proportions of oracle features are not sufficiently comprehensive. Therefore, I recommend a borderline reject.

---

> ### Author Rebuttal · Authors · 2025-07-31
>
> #### *[W1] The KPI framework proposed in the paper does not account for potential noise in the dataset, yet noise is prevalent in industrial scenarios and other real-world applications, which may limit KPI's applicability in such contexts.*
> **Response.** We appreciate the reviewer’s insightful observation regarding the role of noise in real-world datasets and its implications to KPI. We address this concern with a rigorous and evidence-based discussion as follows:
> - Firstly, as acknowledged in our limitations,  KPI assumes that observed data are accurate and free from significant noise. **Notably, that this assumption is not unique to KPI but is a foundational premise shared by most MDI methods in the literature**. We restate and emphasize this general assumption in the limitations section.
> - Secondly, the absence of explicit design for noisy data **does not necessarily imply poor robustness to noise**. To assess KPI under noisy conditions, we conducted additional experiments:
>   - Experimental protocol. We fixed the missing data ratio at 0.1 and introduced additive noise by randomly perturbing 10% of the observed entries, with each perturbed value altered by up to ±20% of its original magnitude.
>   - Experimental results. The results, summarized in the table below, demonstrate that KPI consistently achieves the best performance across noise-involved datasets. The results demonstrate that, although KPI is not specifically designed for noisy data, its superior performance is retained in the noisy scenario.
>
>
> | Dataset:| BT | CC | CBV | IS | PK | QB | WQW |
> |:-|-:|-:|-:|-:|-:|-:|-:|
> | CSDIT|0.6896 | 0.9024 |0.8501 | 0.7580 | 0.7001 |0.5579 | 0.7834 |
> | GAIN |0.7801 | 0.8263 |0.7957 | 0.6242 | 0.5258 |0.4736 | 0.7403 |
> | MICE |0.8743 | 1.0403 |1.0521 | 0.9135 | 0.7402 |0.8541 | 1.0602 |
> | MIRACLE|0.5587 | 0.6591 |0.7735 | 3.7138 | 3.8631 |0.6423 | 0.5978 |
> | MISS.F |0.4886 | 0.7359 |0.7267 | 0.5802 | 0.5919 |0.4346 | 0.6718 |
> | MIWAE|0.5782 | 0.7329 |0.8604 | 0.6242 | 0.6107 |0.5308 | 0.6985 |
> | Sinkhorn |0.5260 | 0.4990 |0.5549 | 0.5043 | 0.4782 |nan| 0.6262 |
> | TDM|0.6394 | 0.7565 |0.7637 | 0.6191 | 0.6064 |0.5083 | 0.7354 |
> | NewImp|0.7802 | 0.8465 |0.8145 | 0.7479 | 0.7558 |0.6046 | 0.7649 |
> | KPI|**0.4446** | **0.3672** |**0.3982** | **0.4123** | **0.3401** |**0.2396** | **0.5112** |
>
>
> #### *[W2] The paper defines oracle features as fully observed features. How can we determine which features can serve as oracle features in practical applications?*
> **Response.** Thank you for raising this important point regarding the practical identification of oracle features. In our framework, "oracle features" are operationally defined as those features that are fully observed—i.e., features for which no entries are missing across all samples in the dataset. **This definition is directly verifiable in real-world data.**
>
> - In practical applications, the selection of oracle features is typically straightforward: **one examines the data matrix and identifies features (columns) with complete observations.** For instance, in electronic health records, demographic variables such as `age`, `gender`, and `height` are routinely and reliably collected for all patients, and thus can be designated as oracle features. In industrial monitoring scenarios, sensor readings from robust or easily accessible equipment (e.g., ambient temperature, pressure at main control points) are often fully recorded, whereas measurements from sensors in harsh or inaccessible environments (e.g., temperature at the base of a high-pressure tower) are more prone to missingness due to equipment failure or maintenance constraints. In such cases, the consistently observed features naturally serve as oracle features.
>
> - We recognize that the number of fully observed features may be limited in some domains. However, our empirical results (Section 4.3) show that even a small number of oracle features can suffice for effective application of KPI. Even worse, in cases where no feature is entirely observed, the definition of "oracle features" may be relaxed to include those with very few missingness, which can be solved through suitable imputation prior to KPI process. We will clarify this operational flexibility and its implications in the revised manuscript.
>
>
> #### *[W3] The KPI method relies on oracle features. Can the KPI method perform effectively when there are few oracle features? In the experimental results shown in Figure 3, the KPI method is only compared with the Miss.F method. However, based on the experimental results in Table 1, the Miss.F method is not the suboptimal method.*
>
> **Response.** Thank you for your valuable and actionable comment. We agree that it is important to evaluate our method in the scenario where only a few oracle features are available and to include additional baselines.
>
> **Initially, we selected Miss.F for comparison because it is a strong representative within iterative imputation methods**, despite not being the overall best performer in Table 1. To address your concern, we have **added experiments using NewImp as a new baseline and investigated the case where only a small number of oracle features are accessible.**
> - Experimental protocol. We simulate this setting using CC and IS datasets, varying the number of oracle features (1, 2, 3, 4), where oracles are specified in order from left to right. Missingness is then introduced to the other features, with a fixed missing ratio of 0.1.
> - Experimental results. The table below summarizes the results for each method under varying numbers of oracle features. As the number of oracle features increases, the performance of KPI improves noticeably. These findings demonstrate that KPI is able to leverage the information from oracle features effectively, leading to substantial performance gains as more fully observed features are available.
>
> | Number of oracle features  | 1 |  | 2 |  | 3 |  | 4 |  |
> |---|--|--|--|--|--|--|--|--|
> |  | MAE  | WASS | MAE  | WASS | MAE  | WASS | MAE  | WASS |
> |CC dataset|
> | MISS.F  | 0.820  | 0.231  | 0.648  | 0.149  | 0.646  | 0.149  | 0.644  | 0.147  |
> | NewImp  | 0.620  | 0.130  | 0.530  | 0.115  | 0.560  | 0.120  | 0.540  | 0.112  |
> | KPI| **0.387**  | **0.083**  | **0.370**  | **0.079**  | **0.369**  | **0.074**  | **0.206**  | **0.022**  |
> | IS dataset|
> | MISS.F  | 0.419  | 0.459  | 0.328  | 0.191  | 0.330  | 0.233  | 0.338  | 0.228  |
> | NewImp  | 0.415  | 0.445  | 0.340  | 0.175  | 0.335  | 0.220  | 0.325  | 0.200  |
> | KPI | **0.363** | **0.426**| **0.268** | **0.114**| **0.268**  |  **0.102**| **0.155**  |  **0.061**|
>
> #### *[W4] In Table 1, only the average results under different missing rates are shown, making it impossible to assess the effectiveness of the KPI method under high missing rates. It is recommended to **present the results under different missing rates separately** to better validate the effectiveness of the KPI method.*
>
> **Response.** Thank you very much for your meticulous and actionable suggestion. In submission, relevant content was included in a **seperate supplementary appendix**, which may not have been sufficiently visible. To address this, we have now explicitly referenced the relevant discussions and results:
> - The results under different missing ratios are seperately reported in `Appendix C.2` (Table 5-8 for missing ratio 0.1, 0.2, 0.3, 0.4).
> - We further provide the results given higher missing ratio (0.6) as follow.
>
> |Datasets | BT  | | CC  | | CBV | | IS  |  | PS  |  | QB  |  | WQW |  |
> |-|-|-|-|-|-|-|-|--|-|--|-|--|-|--|
> | | MAE | WASS  | MAE | WASS  | MAE | WASS  | MAE | WASS | MAE | WASS | MAE | WASS | MAE | WASS
> CSDI-T | 0.746 | 1.647 | 0.931 | 3.045 | 0.843 | 4.111 | 0.772 | 15.419 | 0.721 | 10.429 | 0.558 | 21.528 | 0.754 | 4.54
> Miss.D | 0.783 | 1.746 | 1.006 | 3.32  | 0.877 | 4.399 | 0.841 | 16.961 | 0.743 | 11.264 | 0.591 | 22.604 | 0.769 | 4.858
> MICE | 0.744 | 0.231 | 1.009 | 0.983 | 0.987 | 1.785 | 0.79  | 11.091 | 0.531 | 3.224  | 0.698 | 10.544 | 0.955 | 1.684
> GAIN | 0.727 | 0.72  | 1.016 | 2.233 | 1.294 | 5.083 | 1.26  | 24.406 | 0.799 | 7.758  | 1.188 | 53.39  | 1.423 | 7.177
> Sinkhorn | 0.655 | 0.696 | 0.766 | 1.317 | 0.742 | 1.782 | 0.587 | 7.577  | 0.539 | 4.601  | 0.435 | 8.563  | 0.683 | 1.848
> TDM  | 0.736 | 0.766 | 0.81  | 1.426 | 0.808 | 2.221 | 0.714 | 9.394  | 0.692 | 6.313  | 0.562 | 10.234 | 0.75  | 2.199
> MIWAE  | 0.712 | 0.631 | 0.828 | 1.244 | 0.861 | 2.166 | 0.691 | 9.833  | 0.595 | 4.63 | 0.521 | 12.082 | 0.712 | 1.615
> MISS.F | 0.649 | 0.505 | 0.832 | 1.467 | 0.837 | 2.054 | 0.758 | 11.139 | 0.58  | 3.886  | 0.491 | 8.365  | 0.768 | 2.063
> Remasker | **0.503** | 0.503 | 1.898 | 4.303 | 0.732 | 2.427 | 1.328 | 19.88  | 2.761 | 26.829 | 4.265 | 82.597 | **0.537** | **1.455**
> NewImp | 0.75  | 0.807 | 0.826 | 1.491 | 0.818 | 2.317 | 0.759 | 9.904  | 0.75  | 6.624  | 0.6 | 11.45  | 0.765 | 2.264
> KPI  | 0.598 | **0.435** | **0.557** | **0.841** | **0.732** | **1.585** | **0.517** | **6.835**  | **0.465** | **3.097**  | **0.394** | **7.803**  | 0.708 | 1.533
>
> #### *[W5] In the provided anonymous code link, there is only one file, and its content is ‘okasddsa’.*
>
> **Response.** We sincerely apologize for the previous oversight regarding the anonymous code repository. The issue arose from a miscommunication within our team, which has now been thoroughly addressed. We have since updated the repository to include the complete, fully reproducible implementation of the KPI method, including all necessary scripts for data preprocessing, model training, and evaluation. Additionally, we have included a comprehensive README file that provides detailed instructions on the installation of dependencies and execution of experiments. Once again, we apologize for inconvenience this may have caused.

---

> ### Author Response · Authors · 2025-08-05
> **We provide additional experiments to address W3: performance given a little oracle features**
>
> Dear Reviewer dCSM,
>
> Thank you once again for your constructive and comprehensive feedback!  **In this window, we would like to add results to further address the specific issue [W3]** which we find helpful to strengthen this paper.
>
> ---
>
> #### *[W3] The KPI method relies on oracle features. Can the KPI method perform effectively when there are few oracle features? In the experimental results shown in Figure 3, the KPI method is only compared with the Miss.F method. However, based on the experimental results in Table 1, the Miss.F method is not the suboptimal method.*
>
> **Response.** We would like to further address this concern with **additional experiments**. The experiment results showcase **the effectiveness of KPI given a small amount of oracle features.**  Moreover, **KPI outperforms NewImp across varying settings**, a strong generative method as the additional baseline for comparison.
> - **Firstly, we demonstrate the efficacy of KPI given a few oracle features on different datasets:** CC, IS, BT, and WQW. Overall, **KPI outperforms Miss.F and the newly added NewImp** in different settings. Meanwhile, **KPI can effectively exploit oracle features** for improving performance: as the number of oracle features increases, the performance of KPI improves.
>
> | Number of oracle features  | 1 |  | 2 |  | 3 |  | 4 |  |
> |---|--|--|--|--|--|--|--|--|
> |  | MAE  | WASS | MAE  | WASS | MAE  | WASS | MAE  | WASS |
> | **CC dataset**|
> | MISS.F  | 0.820  | 0.231  | 0.648  | 0.149  | 0.646  | 0.149  | 0.644  | 0.147  |
> | NewImp  | 0.620  | 0.130  | 0.530  | 0.115  | 0.560  | 0.120  | 0.540  | 0.112  |
> | KPI| 0.387  | 0.083  | 0.370  | 0.079  | 0.369  | 0.074  | **0.206**  | **0.022**  |
> | **IS dataset**|
> | MISS.F  | 0.419  | 0.459  | 0.328  | 0.191  | 0.330  | 0.233  | 0.338  | 0.228  |
> | NewImp  | 0.415  | 0.445  | 0.340  | 0.175  | 0.335  | 0.220  | 0.325  | 0.200  |
> | KPI | 0.363 | 0.426| 0.268 | 0.114| 0.268  |  0.102| **0.155**  |  **0.061**|
> | **BT dataset** |
> |      | MAE | WASS | MAE | WASS | MAE | WASS | MAE | WASS |
> | MISS.F | 0.535 | 0.154 | 0.513 | 0.145 | 0.495 | 0.234 | 0.462 | 0.125 |
> | NewImp | 0.502 | 0.148 | 0.521 | 0.149| 0.457 | 0.122 | 0.491 | 0.144|
> | KPI (Ours) | 0.465 | 0.104 | 0.410 | 0.091 | 0.359 | 0.076 | **0.329** | **0.041** |
> | **WQW dataset** |  |
> | MISS.F | 0.809 | 0.160 | 0.769 | 0.150 | 0.741 | 0.137 | 0.758 | 0.147 |
> | NewImp | 0.805| 0.165 | 0.756 | 0.142 | 0.732 | 0.140 | 0.767 | 0.155 |
> | KPI (Ours) | 0.802 | 0.055 | 0.579 | 0.052 | 0.497 | 0.046 | **0.472** | **0.044** |
>
> - **Secondly, we compare the  efficacy of KPI given a few oracle features on different missing ratios:** 0.1, 0.2, and 0.3. Similarly, on the CC dataset, **KPI outperforms Miss.F and NewImp**, and the **performance improves as the number of oracle features increases.** Therefore, KPI exhibits competitive performance given different missig ratios and oracle feature amounts, and can effectively utilize oracle feature for performance improvement.
>
> | **Missing ratio: 0.1**    | 1     |        | 2     |        | 3     |        | 4     |        |
> |-----------|--------|--------|--------|--------|--------|--------|--------|--------|
> |           | MAE    | WASS   | MAE    | WASS   | MAE    | WASS   | MAE    | WASS   |
> | MISS.F    | 0.820  | 0.231  | 0.648  | 0.149  | 0.646  | 0.149  | 0.644  | 0.147  |
> | NewImp    | 0.620  | 0.130  | 0.530  | 0.115  | 0.560  | 0.120  | 0.540  | 0.112  |
> | KPI (Ours) | 0.387  | 0.083  | 0.370  | 0.079  | 0.369  | 0.074  | **0.206**  | **0.022**  |
> | **Missing ratio: 0.2**|
> | MISS.F   | 0.652   | 0.153   | 0.666   | 0.174   | 0.641   | 0.152   | 0.639   | 0.156   |
> | NewImp   | 0.648   | 0.149   | 0.592   | 0.162   | 0.634   | 0.155   | 0.587   | 0.148   |
> | KPI (Ours)| 0.578   | 0.165   | 0.329   | 0.061   | 0.272   | 0.036   | **0.186**   | **0.028**   |
> | **Missing ratio: 0.3**|
> | MISS.F | 0.720 | 0.210 | 0.704 | 0.201 | 0.691 | 0.186 | 0.685 | 0.180 |
> | NewImp | 0.677 | 0.198 | 0.672 | 0.195 | 0.675 | 0.199 | 0.659 | 0.188 |
> | KPI (Ours) | 0.613 | 0.138 | 0.356 | 0.069 | 0.330 | 0.064 | **0.215** | **0.037** |
>
> ---
> We hope that the additional experiments in our rebuttal and discussion phases could resolve the raised issues. Your comment on the oracle features was especially insightful, and the resulting discussion was instrumental in improving our work. **We are very grateful for that valuable guidance.**
>
> Should you find our responses and revisions satisfactory, we would be very appreciative if you would generously adjust your score. If you have any further question and concern, we are happy to discuss.
>
> Many thanks,
>
> Submission28751 Authors

---

> > ### Comment · Reviewer_dCSM · 2025-08-06
> > **Thanks.**
> >
> > I thank the authors for their detailed and thoughtful rebuttal. The additional experiments and clarifications are appreciated and help address several of my initial concerns. However, I maintain my original assessment and rating for the following reasons:
> >
> > While the authors conducted experiments varying the number of oracle features and demonstrated performance improvements, I still find it difficult to fully distinguish the roles of oracle vs. non-oracle features in the model.
> >
> > The method fundamentally relies on the presence of fully observed oracle features.
> >
> > Although the authors have now corrected the code link, the initial failure to provide a working repository raises concerns about reproducibility and the review process timeline. Namely, I have no time to check the correctness of the implementation. I acknowledge the fix but consider this a non-trivial lapse, especially when reproducibility is increasingly emphasized in ML conferences.

---

> > > ### Author Response · Authors · 2025-08-09
> > >
> > > Dear reviewer dCSM,
> > >
> > > Since the discussion period will end in a few hours, we will be online waiting for your feedback on our rebuttal, which we believe has fully addressed your concerns.
> > >
> > > We would highly appreciate it if you could take into account our response when updating the final rating and having discussions with AC and other reviewers.
> > >
> > > Thank you so much for your time and efforts. Sorry for our repetitive messages, but we're eager to ensure everything is addressed.
> > >
> > > 28751 Authors

---

> ### Author Response · Authors · 2025-08-06
>
> Thank you for your detailed feedback and we are delighted that we have cleared your previous concerns!
>
> ---
>
> > **The method fundamentally relies on the presence of fully observed oracle features.**
>
> Thank you for the comment. We would like to clarify that **KPI does not necessarily rely on the presence of oracle features.**
>
> - **Firstly, we agree that KPI excels at exploiting oracle features for improving imputation performance.** It is because KPI incorporates oracle features as supervision signal (when they are available), which provides high-quality signals and therefore beneficial to imputation performance.
>
> - **However, we emphasize that exploitation $\neq$ reliance**: Indeed, **KPI does not necessarily rely on the presence of oracle features.** In implementation, we iteratively treat **each** feature as the target feature $\mathbf{Y}$ to provide supervision, and the remaining as the input features (see line 180). This process is repeated for all features, and does not rely on oracle feature. When there exists oracle features, KPI naturally involves them as part of the supervision signal. When there exists no oracle features, KPI still operates effectively.
>
> - **Experiments further demonstrate KPI does not rely on oracle features.** In `Section 4.2: overall performance`, no oracle feature exists in the generated datasets, but KPI still achieves the best performance. Indeed, the performance given oracle feature presence is separately investigated in `Section 4.3`, where we deliberately controlled the simulation to ensure the existence of oracle features.
>
> > **While the authors conducted experiments varying the number of oracle features and demonstrated performance improvements, I still find it **difficult to fully distinguish the roles of oracle vs. non-oracle features** in the model.**
>
> Thank you for raising this concern. Please kindly note that **KPI does not require distinguishing oracle vs. non-oracle features in the model.**
>
> - In loss function calculation (see line 180), each feature (oracle or non-oracle) is treated as the target feature $\mathbf{Y}$ in a round-robin manner. Even if there exists no oracle features, KPI still works.
>
> - Despite the above, **KPI has a special advantage in exploiting oracle features** in the cases where oracle features are available. The rationale is: oracle feature, as a fruitful information source, explicitly acts as supervision signal (target) to update other feature imputations. This claim is empirically validated in `Section 4.3: oracle feature presence` and the added experiments in rebuttal, where increasing the number of oracle features effectively boosts the performance of KPI.
>
> > **The initial failure to provide a working repository raises concerns about reproducibility and the review process timeline. Namely, I have no time to check the correctness of the implementation. I acknowledge the fix but consider this a non-trivial lapse, especially when reproducibility is increasingly emphasized in ML conferences.**
>
> We are also truly sorry about the accidental problem with the initial link, and during rebuttal have taken immediate action to correct this. The full code is now accessible. To facilitate your checking, we **(i) newly added detailed comments in the code, where we explicitly label the corresponding equations in the paper**, and **(ii) newly included a Dockerfile to automate environment setup and facilitate reproduction.**
>
> In light of the increasing importance of reproducibility in the machine learning community, we commit to releasing the full code for our method and baselines to benefit future research. We hope these actions may address your remaining concerns.
>
> Please also kindly note that as par in NeurIPS 2025 reviewing checklist, it says:
>
> > 5.Open Access to Data and Code: If you ran experiments, did you include the code, data, and instructions needed to reproduce the main experimental results (either in the supplemental material or as a URL)? Please see the NeurIPS code and data submission guidelines for more details. While we encourage release of code and data, we understand that this might not be possible, so no is an acceptable answer. **Papers cannot be rejected simply for not including code, unless this is central to the contribution  (e.g., for a new open-source benchmark).**
>
> Therefore, with the link corrected, we believe further evaluation and discussion can be grounded on our technical contributions.
>
> ***
> Please let us know if we have resolved your concerns – thank you!
>
> Sincerely,
>
> Submission28751 Authors

---

> > ### Author Response · Authors · 2025-08-08
> >
> > Dear Reviewer dCSM,
> >
> > As the discussion deadline approaches, we are wondering whether our responses have properly addressed your concerns? Your feedback would be extremely helpful to us. If you have further comments or questions, we hope for the opportunity to respond to them.
> >
> > Many thanks,
> >
> > 28751 Authors

---

### Official Review · Reviewer_tbrx · 2025-07-02

**Clarity:** 2
**Significance:** 3
**Originality:** 3
**Rating:** 5
**Confidence:** 5

**Summary:**

The paper presents an iterative imputation strategy based on bi-level optimization, proposed as an alternative to the existing MDI method. The approach consists of three steps: (1) randomly select a feature, (2) find a function in a Reproducing Kernel Hilbert Space (RKHS) that accurately predicts this feature using the others, and (3) update the imputation of other features to minimize the prediction risk associated with the selected feature. The method leverages oracle features—those with no missing data—and is empirically validated on a large set of datasets with artificially introduced missing values.

**Questions:**

You could address the points raised under 'Weaknesses' to improve your rating.

Overall, the theoretical overstatements and the omission of the missing data mechanism are the main reasons for my current recommendation to bordeline reject the paper . However, these two issues are relatively easy to address, and given the novelty and practical interest of the method, my evaluation could be revised to "accept" if they are properly corrected.

**Ethical Concerns:**

["NO or VERY MINOR ethics concerns only"]

**Final Justification:**

After rebuttal period the two main concerns are well addressed by the authors: rate: 3->5

**Limitations:**

yes

**Quality:**

3

**Strengths And Weaknesses:**

### **Strengths**

* The method is built upon a well-presented heuristic and offers an interesting alternative for practical applications.
* The imputation literature is well-situated and contextualized.
* The use of RKHS is clearly introduced, and the derivation of its solution is well explained.
* A wide variety of datasets are used for empirical evaluation, which supports the general relevance of the method—although the study remains incomplete in some aspects (see weaknesses).

### **Weaknesses**

* Some of the theoretical results (in main) are incorrect or imprecise. In particular, key assumptions are missing or improperly stated—for instance, the Gaussian kernel is not convex in general, yet convexity is assumed without justification. There are overstatements in the theoretical justifications (e.g., convergence and rates are claimed without sufficient assumptions or guarantees). Moreover, the algorithm uses an approximate minimizer, not the true one, and the true minimization problem is non-convex.
* The paper does not discuss the mechanism of missing data. The proposed method appears to be valid only under the MCAR assumption, which is the most favorable case. Experiments are conducted under this assumption, limiting the scope of comparison with methods that perform well under MAR  or MNAR .
* The method seems to be more of a **completion** strategy than a true **imputation** method. It is unclear how the method could be used to impute missing values on *new* data points without retraining the model, unlike other methods such as MICE or MissForest. The experimental section should include a scenario with a held-out test dataset to evaluate generalization.
* Notation is sometimes imprecise or not rigorous. For example, the function $f^\star$ is applied to inputs that may contain Na values, which is not clearly defined or explained.

Typos (no response needed):
* $X^{imp}$ instead of $X^{(\mathrm{imp})}$ (177)
* Nan (Not a number )-> Na (not attribute)

---

> ### Author Rebuttal · Authors · 2025-07-30
>
> Thank you very much for your constructive comments and appreciation of our heuristic and empirical studies. We are grateful that you have highlighted the key issues and expressed openness to rating adjustment. Below are our responses to the specific concerns and queries.
>
> ---
>
> #### *[W1] Some of the theoretical results are incorrect or imprecise. In particular, key assumptions are missing or improperly stated—for instance, the Gaussian kernel is not convex in general. There are overstatements in the theoretical justifications (e.g., convergence and rates are claimed without sufficient assumptions or guarantees). The algorithm uses an approximate minimizer, not the true one, and the true minimization problem is non-convex.*
>
> **Response.** Thank you for your insightful comments on the theoretical analysis. We acknowledge the raised points and will revise our analysis accordingly.
> - On the convexity of kernel function. In the main paper, we assumed kernel convexity. For Gaussian kernel we demonstrated that it has a convex region given dimension $\mathrm{D}=1$, but it is not convex in general cases with $\mathrm{D}>1$. We acknowledge this imprecision and will now restrict our discussion to the linear kernel, which is convex.
> - On the overstatements like convergence assumptions. Given linear kernel, we immediately have the seperate convexity of the loss function to $X_t$ (since it is a standard quadratic function). Under these conditions, we can demonstrate convergence for the minimization process by updating $X_t$.
> > Considering the learning objective in (8) with linear kernel, define $g(\mathbf{X}^t) = (\mathbf{Y}^t - \mathbf{X}^t \mathbf{w})^2$ where $\mathbf{w} = \mathbf{X}^{s\top}(\mathbf{X}^s\mathbf{X}^{s\top} + \lambda\mathbf{I})^{-1}\mathbf{Y}^s$. Suppose $\mathbf{X}^t_k$ is the $k$-th gradient descent iteration to minimize $g(\mathbf{X}^t)$, $\mathbf{X}^t_*$ is the minimizer of $g(\mathbf{X}^t)$. We make assumtions as follows:
> <!- > - (1) $g$ is $L$-smooth in $\mathbf{X}^t$, i.e., for all $\mathbf{X}, \mathbf{Z}$ we have $\|\nabla g(\mathbf{X}) - \nabla g(\mathbf{Z})\|_F \leq L \|\mathbf{X} - \mathbf{Z}\|_F$;->
> > - (1) $g$ is $L$-strongly convex in $\mathbf{X}^t$, i.e., for all $\mathbf{X}, \mathbf{Z}$ we have $g(\mathbf{Z}) \geq g(\mathbf{X}) + \langle \nabla g(\mathbf{X}), \mathbf{Z} - \mathbf{X} \rangle + \frac{L}{2}\|\mathbf{Z} - \mathbf{X}\|_F^2$
> > - (2) the update step size is small, i.e., $\eta \leq \frac{2}{L}$.
> > - (3) the initial value $\mathbf{X}^t_0$ is proximal to $\mathbf{X}^t_*$, i.e., $\|\mathbf{X}^t_0-\mathbf{X}^t_*\|_2^2 < \epsilon^2$.
>
> > Then, for any $\mathrm{K} \geq 0$, we have$$g(\mathbf{X}^t_\mathrm{K})-g(\mathbf{X}^t_*)\leq \frac{1}{2\eta\mathrm{K}}\epsilon^2.$$
>
>
> > Proof. We first consider the iterative update using gradient descent:$$\mathbf{X}^t_{k+1} = \mathbf{X}^t_k - \eta \nabla_{\mathbf{X}^t} g(\mathbf{X}^t_k).$$ For a $L$-strongly convex function $g$, we derive the **core equation**:\begin{equation}g(\mathbf{X}^t_{k+1})\leq g(\mathbf{X}^t_k)+\nabla_{\mathbf{X}^{t}}^t g(\mathbf{X}^t_k)(\mathbf{X}^t_{k+1}-\mathbf{X}^t_{k})+\frac{L}{2}\|\mathbf{X}^t_{k+1}-\mathbf{X}^t_{k}\|^2\\ =g(\mathbf{X}^t_k)-\eta\|\nabla_{\mathbf{X}^{t}}^tg(\mathbf{X}^t_k)\|^2+\frac{L\eta^2}{2}\|\nabla_{\mathbf{X}^{t}}^tg(\mathbf{X}^t_k)\|^2\\ \leq g(\mathbf{X}^t_k)-\frac{\eta}{2}\|\nabla_{\mathbf{X}^{t}}^tg(\mathbf{X}^t_k)\|^2 \\ \leq g(\mathbf{X}^t_*)+\nabla_{\mathbf{X}^{t}}^tg(\mathbf{X}^t_k)(\mathbf{X}^t_k-\mathbf{X}^t_*)-\frac{\eta}{2}\|\nabla_{\mathbf{X}^{t}}^tg(\mathbf{X}^t_k)\|^2 \\ = g(\mathbf{X}^t_*)+\frac{1}{2\eta}\|\mathbf{X}^t_k-\mathbf{X}^t_*\|^2 - \frac{1}{2\eta}\|\mathbf{X}^t_k-\mathbf{X}^t_*\|^2 +\nabla_{\mathbf{X}^{t}}^tg(\mathbf{X}^t_k)(\mathbf{X}^t_k-\mathbf{X}^t_*)-\frac{\eta}{2}\|\nabla_{\mathbf{X}^{t}}^tg(\mathbf{X}^t_k)\|^2\\ = g(\mathbf{X}^t_*)+\frac{1}{2\eta}\|\mathbf{X}^t_k-\mathbf{X}^t_*\|^2 - \frac{1}{2\eta}\|\mathbf{X}^t_k-\mathbf{X}^t_*-\eta\nabla_{\mathbf{X}^{t}}^tg(\mathbf{X}^t_k)\|^2\\ = g(\mathbf{X}^t_*)+\frac{1}{2\eta}(\|\mathbf{X}^t_k-\mathbf{X}^t_*\|^2 - \|\mathbf{X}^t_{k+1}-\mathbf{X}^t_*\|^2),\end{equation} where the first equation uses assumption 1; the second equation uses the update rules of the gradient descent; the fourth step holds as we set $\eta \leq \frac{1}{L}$ which makes $- \eta + \frac{L \eta^2}{2} \leq -\frac{\eta}{2}$.
>
> > Summing up the inequality above with $k=0,1,...,\mathrm{K}$, we have: \begin{equation} \sum_{k=1}^\mathrm{K} g(\mathbf{X}^t_{k}) - \mathrm{K}g(\mathbf{X}^t_*) \leq \frac{1}{2\eta}(\|\mathbf{X}^t_0-\mathbf{X}^t_*)\|^2-\|\mathbf{X}^t_\mathrm{K}-\mathbf{X}^t_*\|^2)\leq \frac{1}{2\eta}\|\mathbf{X}^t_0-\mathbf{X}^t_*\|^2.\end{equation} According to the 4-th step of the core equation, we have $g(\mathbf{X}^t_{k+1})\leq g(\mathbf{X}^t_{k})\leq...\leq g(\mathbf{X}^t_{0})$. Therefore, $\mathrm{K}g(\mathbf{X}^t_{\mathrm{K}})\leq\sum_{k=1}^\mathrm{K}g(\mathbf{X}^t_{k})$ holds, which immediately follows by\begin{equation} \mathrm{K}g(\mathbf{X}^t_\mathrm{K})-\mathrm{K}g(\mathbf{X}^t_*)\leq\sum_{k=1}^\mathrm{K} g(\mathbf{X}^t_{k}) - \mathrm{K}g(\mathbf{X}^t_*)\leq \frac{1}{2\eta}\|\mathbf{X}^t_0-\mathbf{X}^t_*\|^2\leq\frac{1}{2\eta}\epsilon^2. \end{equation} Therefore, we have: \begin{equation} g(\mathbf{X}^t_\mathrm{K})-g(\mathbf{X}^t_*)\leq \frac{1}{2\eta\mathrm{K}}\epsilon^2.\end{equation}
> - On the approximate minimizer. We agree that the convergence justification relied on stringent assumptions (e.g., Gaussian kernel and separate convexity), and in practice only targets the approximate minimizer, which largely weakens its significance. In our revised manuscript, we will remove discussions on algorithmic convergence. Instead, we will limit the discussion in the appendix to the separate convexity of the objective under the linear kernel, thereby ensuring precision and avoiding overstatement.
>
>
>
> #### *[W2] The paper does not discuss the mechanism of missing data, limiting the scope of comparison with methods that perform well under MAR or MNAR.*
> **Response.** Thank you very much for your meticulous and actionable suggestion. In submission, relevant content was included in a **seperate supplementary appendix**, which may not have been sufficiently visible. To address this, we have now explicitly referenced the relevant discussions and results:
> - The MAR and MNAR mechanisms are discussed in `Appendix B.1`.
> - The MAR and MNAR results are discussed in `Appendix C.3` with missing ratio fixed as 0.1.
>
> #### *[W3] The method seems to be more of a completion strategy than a true imputation method. It is unclear how the method could be used to impute missing values on new data points without retraining the model, unlike other methods such as MICE or Miss.F. The experimental section should include a scenario with a held-out test dataset to evaluate generalization.*
>
> **Response.** Thank you for your insightful comment. Please see our concise response below.
> - Firstly, We agree that our method is best characterized as a **completion strategy**, similar to prior works [1,2]. To clarify this, we will revise the terminology throughout the manuscript to `completion` where appropriate. In particular, we will add the following statement to the preliminary note (line 58):
> > As a preliminary note, there are two primary paradigms for handling missing data: (1) completion, where the model is optimized to fill in missing entries within a given incomplete dataset and does not generalize to new missing indices; and (2) imputation, where the model is trained to impute missing values for previously unseen indices or samples. The completion setting is practical in contexts where the entire dataset is collected prior to analysis. Consistent with prior work such as Sinkhorn and TDM [1,2], this study adopts the completion paradigm.
> - Secondly, as more of a completion approach, the current method does not naturally generalize to imputing missing values in new data points without training.
> - Finally, we are pleased that **it is feasible to add the requested experiment using a held-out test dataset to evaluate how well KPI generalizes to new samples**. Specifically, for each new sample, KPI minimizes the learning objective (Eq. 8) via gradient descent, updating only the values on missing indices in the test sample while keeping those in the training data fixed. The results in the table below demonstrate that KPI retains effective in the held-out test setting.
>
> | Dataset  |CC| | IS| |
> |---|-|--|---|--|
> |    | MAE   | WASS  | MAE   | WASS  |
> | **MICE**     | 0.258 | 0.045     | 0.544 | 0.737     |
> | **MISS.F**     |0.222|0.040| 0.194 | 0.085     |
> | **KPI**  | 0.107 | 0.054     | 0.107 | 0.069     |
>
> **Reference**
>
> [1] Missing data imputation using optimal transport. ICML.
>
> [2] Transformed distribution matching for missing value imputation. ICML.
>
> #### *[W4] Notation is sometimes imprecise or not rigorous. For example, the function is applied to inputs that may contain Na values, which is not clearly defined or explained.*
> **Response.** Thank you for highlighting the notational clarity. In Eq. 4, we formulated $f^* (\mathbf{X}^\mathrm{(miss)}\_{\cdot,-d},\mathbf{X}^\mathrm{(obs)}\_{\cdot,-d})$.
> As you noted, it is ambiguous to apply $f
> ^*$ to inputs with NaN values. Nevertheless, in practice, as you could have noticed, **the missing entries are initialized with real values prior to optimization**, so $f$ only receives **real-valued inputs** during the KPI process. To avoid confusion, we will replace $\mathbf{X}^\mathrm{(miss)}$ with $\mathbf{X}^\mathrm{(comp)}$ —explicitly denoting completed values—in this and related expressions. Other notations will similarly be standarized to ensure rigor and consistency.

---

> ### Author Response · Authors · 2025-08-04
> **Please find our additional clarifications and experiments for the two main concerns [1/2]**
>
> Dear Reviewer tbrx,
>
> We noticed the two concerns highlighted in the review report. In this thread, we would like to offer supplementary responses and results to further clarify these points.
>
> #### **[Additional response to W1, convergence].**
> **Response.** We acknowledged the overstatement and would like to add experiments to demonstrate convergence.
> - **Revising Claims of Convergence.**  We concur with the reviewer that our algorithm utilizes an approximate minimizer for a non-convex optimization problem when general kernels are used. To rectify our previous overstatement, we would remove the theoretical claims regarding algorithmic convergence. The discussion in the appendix will be confined to the separate convexity of the loss function under the linear kernel, which avoids this overstatement.
> - **Adding Empirical Demonstration of Convergence.** While a theoretical proof of convergence remains challenging, we provide empirical results that demonstrate the convergence of our algorithm in practice. Specifically, in the tables below, we report the training loss, the validation MAE, and Wasserstein discrepancy at different epochs on CC dataset. Three key observations are noted.
>     - The training loss  decreases as the number of epochs increases, which indicates that the **SGD-based optimizer is effective at minimizing the loss function.**
>     - The decrease in the training loss corresponds to a reduction in the validation set error, confirming that **minimizing the loss function is effective to improve imputation accuracy.**
>     - The training loss and validation metrics stabilize after a certain number of epochs, indicating that **the optimization process has converged.**
>
> **MCAR results with missing ratio 0.1**
> |    |      |       | CC  |       |
> |:--:|------|:-----:|:-----:|-------|
> |    | Loss | MAE   | WASS  | MSE   |
> | 0  |    0.582  | 0.845 | 0.572 | 1.027 |
> | 20 |  0.461    | 0.120 | 0.152 | 0.218 |
> | 40 |   0.435  | 0.098 | 0.044 | 0.072 |
> | 60 |    0.422  | 0.090 | 0.038 | 0.064 |
>
> **MAR results with missing ratio 0.1**
> |    |      |       |   CC  |       |
> |:--:|------|:-----:|:-----:|-------|
> |    | Loss | MAE   | WASS  | MSE   |
> | 0  |    0.521  | 0.887 | 0.627 | 1.031 |
> | 20 |  0.282    | 0.073 | 0.045 | 0.062 |
> | 40 |   0.271  | 0.068 | 0.038 | 0.059 |
> | 60 |   0.269   | 0.065 | 0.036 | 0.058 |
>
> **MNAR results with missing ratio 0.1**
>
> |    |      |       |   CC  |       |
> |:--:|------|:-----:|:-----:|-------|
> |    | Loss | MAE   | WASS  | MSE   |
> | 0  |    0.497  | 0.829 | 0.627 | 1.031 |
> | 20 |   0.445   | 0.325 | 0.228 | 0.376 |
> | 40 |    0.411  | 0.180 | 0.115 | 0.195 |
> | 60 |   0.403   | 0.158 | 0.093 | 0.163 |
>
>
> #### **[Additional response to W2, MNAR and MAR results].**
> **Response.** We would like to address this concern by (1) discussing the MAR and MNAR mechanisms, and (2) adding experiment results.
> - **MAR and MNAR merchanism.** In this paper, **to simulate MAR data**, features are randomly divided into two categories: observable and missing. Missingness is determined by a logistic model influenced by observable features, parameterized by randomly selected weights and a bias adjusted to achieve the desired missing ratio. **To simulate MNAR data**, the non-missing features from the MAR simulation are performed with a secondary MCAR masking. This additional step creates dependencies between missing values and certain observable features, thus replicating the MNAR condition.
> - **MAR and MNAR results.** We investigated the performance of baselines and KPI given MAR and MNAR scenarios. The results can be found in the supplementary Appendix file, for a missing ratio of 0.1. Please kindly see Table 9 for MAR results and Tabel 10 for MNAR results.
> - **Additional MAR and MNAR results.** We understand that the reviewer may advocate for more comprehensive results in MAR and MNAR scenarios. Therefore, **we have added experiments with an increased missing ratio of 0.3 for both MAR and MNAR conditions.** The results are presented in the tables below. Similar to the MCAR results, KPI exhibits competitive performance across most datasets, showing its robustness to different missing data mechanisms. These results will be collectively presented in the supplementary Appendix.

---

> ### Author Response · Authors · 2025-08-04
> **Please find our additional clarifications and experiments for the two main concerns [2/2]**
>
> | Method | | BT| | CC| | CBV | | IS| | PK| | QB| | WQW| |
> |-----------|-------|--------|-------|--------|-------|--------|-------|--------|-------|--------|-------|--------|-------|---------|-------|
> | | | MAE | WASS | MAE | WASS | MAE | WASS | MAE | WASS | MAE | WASS | MAE | WASS | MAE| WASS |
> | MICE | | 0.589| 1.362 | 0.733| 0.695 | 0.945| 1.637| 0.771| 2.798| 0.599| 2.083| 0.697| 4.563| 0.917 | 0.962 |
> | MISS.F| | 0.825| 0.815 | 0.739| 0.517 | 0.876| 0.659| 0.673| 1.338| 0.903| 1.604| 0.685| 2.802| 0.814 | 0.745 |
> | CSDI_T| | 1.187| 5.632 | 1.002| 3.275 | 0.934| 4.412 | 0.812| 16.143| 1.355| 19.178| 0.891| 24.231| 0.921 | 5.084 |
> | MissDiff| | 1.125| 2.982 | 0.995| 2.317 | 0.947| 6.063 | 0.798| 14.312| 1.327| 23.845| 0.869| 35.042| 0.912 | 6.953 |
> | GAIN| | 1.175| 1.342 | 0.891| 0.705 | 0.793| 0.587 | 0.563| 0.734 | 0.816| 1.526| 0.645| 1.879| 0.837 | 0.805 |
> | MIRACLE | | 0.804| 0.653 | 0.462| 0.236 | 0.817| 0.645 | 3.942| 20.915| 4.625| 24.791| 0.698| 1.781| 0.602 | 0.472 |
> | MIWAE | | 0.852| 0.921 | 0.829| 0.604 | 0.895| 0.732 | 0.568| 0.897 | 0.864| 1.759| 0.755| 3.021| 0.735 | 0.662 |
> | Sinkhorn| | 1.213| 1.453 | 1.065| 0.843 | 1.017| 0.891 | 0.935| 1.487 | 1.364| 3.521| 1.011| 3.185| 1.012 | 1.028 |
> | TDM | | 1.108| 1.396 | 0.962| 0.729 | 0.918| 0.768 | 0.786| 1.123 | 1.302| 3.378| 0.847| 2.814| 0.901 | 0.891 |
> | ReMasker| | 0.526| 0.524 | 0.602| 0.435 | 0.491| 0.376 | 0.628| 2.239 | 0.612| 1.597 | 0.607| 3.813 | 0.519 | 0.591 |
> | NewImp| | 0.527| 0.382 | 0.357| 0.257 | 0.389| 0.330 | 0.465| 1.564 | 0.446| **1.213** | 0.451| 3.509 | 0.468 | 0.556 |
> | KPI (ours)| | **0.394**|**0.160**|**0.153**|**0.129**|**0.288**|**0.238**|**0.364**|**1.396**|**0.437**|1.420|**0.295**|**3.115**|**0.423**|**0.476**|
>
> | Method | BT || CC || CBV|| IS || PK || QB || WQW||
> |------------|--------|--------|--------|--------|--------|--------|--------|--------|--------|--------|--------|--------|--------|--------|
> || MAE| WASS | MAE| WASS | MAE| WASS | MAE| WASS | MAE| WASS | MAE| WASS | MAE| WASS |
> | MICE | 0.696| 0.411| 0.660| 0.384| 0.832| 0.734| 0.708| 1.831| 0.499| 0.980| 0.589| 2.377| 0.807| 0.724|
> | MISS.F | 0.731| 0.579| 0.697| 0.483| 0.748| 0.631| 0.587| 1.944| 0.726| 1.513| 0.500| 2.291| 0.667| 0.567|
> | CSDI-T | 0.885| 3.105| 0.885| 2.923| 0.838| 3.922| 0.759| 16.833 | 1.016| 14.173 | 0.683| 20.330 | 0.795| 4.275|
> | MissDiff | 0.912 | 2.473 | 0.854 | 1.193 | 0.842 | 3.704 | 0.752 | 13.391 | 0.991 | 19.323 | 0.662 | 23.292 | 0.771 | 5.953 |
> | GAIN | 0.846| 0.595| 0.782| 0.545| 0.698| 0.570| 0.529| 1.511| 0.571| 1.305| 0.474|**1.943**| 0.730| 0.684|
> | MIRACLE| 0.655| 0.319| 0.371| 0.163| 0.842| 0.802| 3.725| 27.093 | 4.196| 25.052 | 0.576| 2.249| 0.518| 0.437|
> | MIWAE| 0.658| 0.424| 0.735| 0.508| 0.808| 0.731| 0.559| 1.535| 0.628| 1.400| 0.561| 3.318| 0.641| 0.577|
> | Sinkhorn | 0.967| 0.752| 0.940| 0.698| 0.925| 0.911| 0.854| 2.121| 1.049| 2.906| 0.844| 3.755| 0.880| 0.859|
> | TDM| 0.858| 0.732| 0.849| 0.620| 0.821| 0.756| 0.724| 1.640| 0.969| 2.713| 0.661| 3.202| 0.772| 0.744|
> | Remasker |0.481| 0.297| 1.028| 0.886| 0.487|**0.296**| 0.660| 1.701| 0.586| 1.059| 0.516| 2.128| 0.519|**0.434**|
> | NewImp | 0.645| 0.461| 0.585| 0.593| 0.562| 0.837| 0.442| 3.945| 0.434| 2.328| 0.441| 7.161| 0.601| 1.102|
> | KPI (ours) |**0.455**|**0.124**|**0.158**|**0.093**|**0.357**|0.352 |**0.391**|**1.344**|**0.423**|**0.769**|**0.286**|1.967 |**0.458**|0.563 |
>
> ---
> We sincerely appreciate your encouraging words that *Overall, the theoretical overstatements and the omission of the missing data mechanism are the main reasons for my current recommendation to bordeline reject the paper. However, these two issues are relatively easy to address, and given the novelty and practical interest of the method, my evaluation could be revised to "accept" if they are properly corrected.* In light of this, if our response and additional experiments have properly resolved your raised issues, we politely invite you to further reconsider your overall score.
>
> I personally really enjoy the discussion with you. I am particularly grateful for your thoughtful approach—not only highlighting crucial points for improvement but also offering encouragement, a practice we also value in the review process. Your comments on toning down overstatements were especially helpful, which makes work much more rigorous as a result.
>
> If you have any further question and concern, we are happy to discuss.
>
> Thanks for your time,
>
> Submission28751 Authors

---

> > ### Comment · Reviewer_tbrx · 2025-08-06
> > **Increase rate to accept**
> >
> > Based on my initial review, my two main concerns have been addressed by the authors. Therefore, I am updating my recommendation to 'accept.' I appreciate the authors' responses and the quality of their work.

---

> ### Author Response · Authors · 2025-08-06
> **Thank you for updating your recommendation to 'accept'**
>
> Thank you so much for updating your recommendation to 'accept' and recognizing our work quality. We highly appreciate your encouraging words and we are truly grateful for your dedicated time and constructive comments!

---

> ### Author Response · Authors · 2025-08-09
>
> Dear reviewer tbrx,
>
> We really appreciate your recognition of our work and your kind words, for considering "the introduction of our heuristic and the use of RKHS are the real strengths", "this paper is a refreshing and inspiring contribution to the imputation literature", "it combines originality with significance", and "it’s highly interesting even without any formal theoretical guarantees" --  thank you for your engagement with our work.
>
> In addition, thank you for your constructive suggestions for toning down claims and incorporating the additional MNAR/MAR experiments are excellent points, and we will be integrating them in our final version.
>
> We are also truly sorry about the accidental problem with the initial link, and during rebuttal have taken immediate action to correct this. The full code is now accessible, especially includes a detailed collection of comments (for easy understanding) and a Dockerfile (for easy replication). Meanwhile, we are reminded of the conference guidelines that leave room for addressing the link issue in the rebuttal phase as follows:
>
> > 5.Open Access to Data and Code: If you ran experiments, did you include the code, data, and instructions needed to reproduce the main experimental results (either in the supplemental material or as a URL)? Please see the NeurIPS code and data submission guidelines for more details. While we encourage release of code and data, we understand that this might not be possible, so no is an acceptable answer. **Papers cannot be rejected simply for not including code, unless this is central to the contribution  (e.g., for a new open-source benchmark).**
>
> Therefore, with the link corrected, we trust the evaluation and discussion can be grounded on our technical contributions (for which you have highly recognized and we are truly encouraged).
>
> Thanks for your time,
>
> Submission28751 Authors

---

### Comment · Area_Chair_kRNu · 2025-08-04

Reviewers, please engage in the discussion with the authors. The discussion period will end in a few days.

---

### Note · Authors · 2025-08-11

Dear AC and all reviewers,

We sincerely appreciate your commitment to evaluating our paper despite your busy schedules. We are encouraged that 3 out of 4 reviewers are currently in the positive side, with scores 5 with conf 5, 4 with conf 4, and 4 with conf 3, recognizing our paper
- "offers an interesting alternative for practical applications" (Reviewer tbrx),
- "cleverly addresses the two core issues in iterative imputation" (Reviewer 8EZS),
- is "a refreshing and inspiring contribution to the imputation literature" (Reviewer tbrx).

At the same time, we noticed that Reviewer dCSM give an initial rating of 3 with conf 3 (no final rating and justification yet). After acknowledging "the additional experiments and clarifications are appreciated and help address several of my initial concerns", dCSM's current concerns are on the (1) roles and reliance on the oracle features, and (2) code availability.

> The method fundamentally relies on the presence of fully observed oracle features.
- We were wondering if there was a misreading - while KPI excels at exploiting oracle features to improve imputation, this is only one of its advantages. In fact, **our method does *not* relies on the presence of fully observed oracle features**, and we provided empirical supports (please kindly refer to our full response).

> The initial failure to provide a working repository raises concerns about reproducibility and the review process timeline. Namely, I have no time to check the correctness of the implementation.
- We sincerely apologize for the accidental problem with the initial link, and have taken immediate action to correct this. The full code is accessible, especially with added comments (for easy understanding) and a Dockerfile (for easy replication).
- We respectfully have reservations about our initial link problem should not be used as a main reason for rejecting our paper. Due to NeurIPS 2025 reviewing policy, it says
> 5.Open Access to Data and Code: If you ran experiments ... **Papers cannot be rejected simply for not including code, unless this is central to the contribution (e.g., for a new open-source benchmark).**

We are confident that our responses can thoroughly address Reviewer dCSM's concerns. We kindly remind that Reviewer dCSM have not replied to our further comments. Thus, we respectfully ask AC and reviewers to consider this context when making your recommendation. Thank you for your invaluable time and effort.

Many thanks,

The Authors

---

### Decision · Program_Chairs · 2025-09-17

**Decision:**

Accept (poster)

**Comment:**

This paper presents a new method that uses an RKHS and oracle features (features with no or few missing attributes) to perform imputation. The experiments are thorough and demonstrate the method's superiority in imputation tasks compared to other methods. The method is also well-principled and has some supporting theory.

Three of the four reviewers recommend or lean towards accept. The fourth reviewer (dCSM) leans toward reject. This reviewer seems to have two lingering concerns. The first is a concern is that the use of oracle features is potentially limiting. Initially, several other reviewers shared this concern. However, the authors subsequently demonstrated robustness of their method to this aspect. Their response ultimately satisfied the other reviewers as well as me.

The second concern was a lack of initial availability of code. However, NeurIPS policy makes this clear that a paper cannot be rejected solely for this, except where the code distribution is central to the paper, which it is not in this case.

Given this, I recommend acceptance of the paper despite the fourth reviewer's concerns.